# A membrane associated tandem kinase from wild emmer wheat confers broad-spectrum resistance to powdery mildew

Miaomiao Li [1,10] ✉, Huaizhi Zhang[1,10], Huixin Xiao[1,10], Keyu Zhu[1,2,10], Wenqi Shi[3,10], Dong Zhang[4], Yong Wang[5], Lijun Yang[3], Qiuhong Wu[1], Jingzhong Xie [1], Yongxing Chen[1,2], Dan Qiu[1], Guanghao Guo[1], Ping Lu[1], Beibei Li[1,2], Lei Dong[1,2], Wenling Li[1,2], Xuejia Cui[1,2], Lingchuan Li[1,2], Xiubin Tian[1], Chengguo Yuan[6], Yiwen Li [1], Dazhao Yu[3], Eviatar Nevo[7], Tzion Fahima [7], Hongjie Li[8], Lingli Dong [1] ✉, Yusheng Zhao [1,2] ✉ & Zhiyong Liu [1,2,9] ✉

Crop wild relatives offer natural variations of disease resistance for crop improvement. Here, we report the isolation of broad-spectrum powdery mildew resistance gene *Pm36*, originated from wild emmer wheat, that encodes a tandem kinase with a transmembrane domain (WTK7-TM) through the combination of map-based cloning, PacBio SMRT long-read genome sequencing, mutagenesis, and transformation. Mutagenesis assay reveals that the two kinase domains and the transmembrane domain of WTK7-TM are critical for the powdery mildew resistance function. Consistently, in vitro phosphorylation assay shows that two kinase domains are indispensable for the kinase activity of WTK7-TM. Haplotype analysis uncovers that *Pm36* is an orphan gene only present in a few wild emmer wheat, indicating its single ancient origin and potential contribution to the current wheat gene pool. Overall, our findings not only provide a powdery mildew resistance gene with great potential in wheat breeding but also sheds light into the mechanism underlying broad-spectrum resistance.

Wheat (*Triticum aestivum* L.) is the world's most widely grown food crop, providing 20% of the total daily calories and protein in human nutrition[1]. Pathogens and pests have been imposing big challenges on wheat production, therefore threatening global food security. Wheat powdery mildew, caused by *Blumeria graminis* f. sp. *tritici* (*Bgt*), is one of the most devastating diseases causing dramatic reductions in both grain yield and quality over the past decades across the world[2].

Improving crop resistance to pathogens through breeding is an effective and environmentally sound strategy for managing disease and minimizing these losses. However, the emergence of new virulent pathogen populations always makes resistant wheat varieties become susceptible[3]. Therefore, resistance breeding is always like an 'arm race' and looking for broad-spectrum and durable disease resistance genes is still needed.

[1]State Key Laboratory of Plant Cell and Chromosome Engineering, Institute of Genetics and Developmental Biology, The Innovative Academy of Seed Design, Chinese Academy of Sciences, Beijing, China. [2]College of Advanced Agricultural Sciences, University of Chinese Academy of Sciences, Beijing, China. [3]Institute of Plant Protection and Soil Science, Hubei Academy of Agricultural Sciences, Wuhan, China. [4]Beijing PlantTech Biotechnology Co., Ltd., Beijing, China. [5]Institute of Vegetables and Flowers, Chinese Academy of Agricultural Sciences, Beijing, China. [6]Hebei Gaoyi Seeds Farm, Gaoyi Hebei, China. [7]Institute of Evolution, University of Haifa, Mt. Carmel Haifa, Israel. [8]National Engineering Laboratory for Crop Molecular Breeding, Institute of Crop Sciences, Chinese Academy of Agricultural Sciences, Beijing, China. [9]Hainan Seed Industry Laboratory, Sanya City, Hainan Province, China. [10]These authors contributed equally: Miaomiao Li, Huaizhi Zhang, Huixin Xiao, Keyu Zhu, Wenqi Shi. ✉e-mail: mmli@genetics.ac.cn; lldong@genetics.ac.cn; yusheng.zhao@genetics.ac.cn; zyliu@genetics.ac.cn

To date, eighteen powdery mildew resistance (*Pm*) genes, *Pm1a*[4], *Pm2*[5], *Pm3*[6], *Pm4*[7], *Pm5e*[8], *Pm8*[9], *Pm12*[10], *Pm13*[11], *Pm17*[12], *Pm21*[13,14], *Pm24*[15], *Pm38/Yr18/Lr34/Sr57*[16], *Pm41*[17], *Pm46/Yr46/Lr67/Sr55*[18], *Pm57*[19], *Pm60*[20], *Pm69*[21] and *WTK4*[22] have been cloned in wheat and its wild relatives. Most of them encode coiled coil nucleotide-binding leucine-rich-repeat (NLR) proteins, except for *Pm4*, *Pm13*, *Pm24*, *WTK4*, *Pm57*, *Pm38/Yr18/Lr34/Sr57*, and *Pm46/Yr46/Lr67/Sr55*. *Pm4* encodes a putative chimeric protein of a serine/threonine kinase, multiple C2 domains, and transmembrane regions that mediate race-specific resistance[7]. *Pm13* from *Aegilops longissimi* encodes a unique mixed lineage kinase domain-like protein[11]. *Pm24* from Chinese wheat landrace Hulutou, *WTK4* from *Aegilops tauschii*, and *Pm57* from *Aegilops searsii* encode tandem kinase protein (TKP)[15,19,22]. The partial resistance genes *Pm38/Yr18/Lr34/Sr57* and *Pm46/Yr46/Lr67/Sr55* encode an ATP-binding cassette (ABC) transporter[16] and a hexose transporter[18], respectively.

Recently, a class of resistance genes encoding TKPs has emerged as new players in plant immunity[23], including barley rust resistance genes *Rpg1*[24]; wheat rust resistance genes *Yr15* (*WTK1*)[25], *Sr60* (*WTK2*)[26], *Sr62* (*WTK5*)[27], and *Lr9* (*WTK6-vWA*)[28]; wheat powdery mildew resistance genes *Pm24* (*WTK3*)[15], *WTK4*[22], and *Pm57*[19], as well as wheat blast resistance gene *Rwt4*[29]. Most of them are composed of putative kinase and pseudokinase domains in tandem, both of which belong to the protein kinase-like superfamily with a conserved catalytic domain; however, none contained neither membrane-targeting motifs nor known receptor sequences[23], leaving a mystery how these TKPs perceive plant immunity signal. More understanding of how TKPs perform their function would boost their application in wheat resistance breeding.

*Pm36*, a wild emmer wheat (WEW)-derived powdery mildew resistance gene, was identified and mapped on chromosome arm 5BL in the tetraploid WEW-durum introgression line 5BIL-29[30]. We previously mapped a powdery mildew resistance gene *Ml3D232*, derived from WEW in the hexaploid wheat introgression line 3D232, on the overlapping genetic interval with *Pm36*[31]. We then deduced that they are likely the same gene or alleles by comparative genetic linkage maps[32]. However, the molecular relationship of the two genes is still unclear.

In this study, we report map-based cloning of *Ml3D232/Pm36*, originated from WEW and confers broad-spectrum resistance to *Bgt* in both durum and common wheat. *Pm36* encodes a unique tandem kinase with a transmembrane domain in the C-terminus (WTK7-TM). We demonstrate that the two kinase domains (Kin I and Kin II) and the transmembrane domain of WTK7-TM are critical for its powdery mildew resistance function. We also show that both Kin I and Kin II are indispensable for the kinase activity of WTK7-TM. *Pm36* is only present in a few WEW populations with a single ancient origin and can potentially contribute to modern wheat improvement.

## Results

### Fine mapping of *Ml3D232*

WEW-hexaploid wheat introgression line 3D232 exhibits a broad-spectrum resistance against an extensive collection of tested 104 genetically distinct *Bgt* isolates at the seedling stage, including the most prevailing virulent isolates E09, E20, and E21 from China (Supplementary Fig. 1 and Supplementary Data 1). 3D232 is also highly resistant to *Bgt* isolate E09 at the adult plant stage under the field condition (Supplementary Fig. 2). The powdery mildew resistance gene *Ml3D232* in 3D232 was previously identified and mapped on chromosome arm 5BL[31]. Using comparative genetic linkage mapping, we found the mapping interval of *Ml3D232* overlapped with *Pm36* derived from WEW, indicating that they are likely the same gene or allelic[32].

To find mapping *Ml3D232* gene, two *Ml3D232*-flanking simple sequence repeats (SSR) markers *XBD37670* and *XBD37760* on

chromosome 5BL[32] were selected to screen 15,893 individual F₂ plants derived from the cross between a highly susceptible common wheat line Xuezao and the resistant introgression line 3D232 (Fig. 1a and Supplementary Data 2). Finally, *Ml3D232* was narrowed down to a genetic interval of 0.021-cM flanked by the closest derived cleaved amplified polymorphic sequence (dCAPS) marker *WGGBM2* and SSR marker *WGGBM7* (Fig. 1b and Supplementary Data 2), which corresponds to 273.8 kb genomic region of wild emmer wheat Zavitan (WEW_v1.0)[33] and contains four predicted high-confidence genes, a cytochrome B561 and domon domain protein (CYB561-Domon), two serpin proteins (*Serpin 1* and *Serpin 2*), and a transmembrane protein (TM9SF4) (Fig. 1c and Supplementary Data 2). qRT-PCR analyses showed that *CYB561-Domon* and *TM9SF4* were constitutively expressed in leaves of 3D232 and Xuezao seedlings, regardless of inoculation with *Bgt* isolate E09, whereas no expression of the two serpin genes was detected in either 3D232 or Xuezao (Supplementary Fig. 3a, b). Since differential expressions and coding sequence, polymorphisms were observed for *CYB561-Domon* (Supplementary Figs. 3 and 4) and *TM9SF4* (Supplementary Figs. 3 and 5) genes between 3D232 and Xuezao, two overexpression constructs, *ProUbi*: *CYB561-Domon* and *ProUbi*: *TM9SF4*, consist of the coding sequences (CDS) of the two genes from 3D232 and driven by the maize (*Zea mays* L.) *Ubiquitin* promoters (Supplementary Fig. 6a, b), were separately delivered into highly susceptible wheat cultivar Fielder by *Agrobacterium*-mediated transformation. Unfortunately, all plants positive for *ProUbi*: *CYB561-Domon* and *ProUbi*: *TM9SF4* constructs in the independent T₁ transgenic families were highly susceptible to the *Bgt* isolate E09 (Supplementary Fig. 6c–e), suggesting *CYB561-Domon* and *TM9SF4* are not the causal gene of *Ml3D232*.

### PacBio SMRT long-read sequencing of tetraploid WEW-durum introgression line 5BIL-29

In order to isolate the causal gene underpinning locus *Ml3D232/Pm36*, we selected the tetraploid WEW-durum introgression line 5BIL-29 carrying *Pm36*[30] for sequencing through the PacBio SMRT long-read sequencing approach. PacBio CCS (HiFi) reads corresponding to approximately 19.4-fold coverage of tetraploid wheat genome were generated (Supplementary Data 3). In addition, about 117.3-fold Hi-C reads were added to the assembly. After sequence assembling, 2207 contigs were obtained, with a contig N50 of 20.8 Mb and N90 of 4.5 Mb, the longest contig length of 168.3 Mb, and the total length of the genome assembly of 10.5 Gb, which covered the whole genome of WEW Zavitan[33] and durum wheat Svevo[34] (Supplementary Data 3).

A 7.1 Mb contig ptg000422 spanning the entire *Pm36* physical mapping interval (1.17 Mb) was captured by aligning the flanking and co-segregating markers *WGGBM2* to *WGGBM7* around the *Pm36* locus (Fig. 1d). There were eleven annotated genes in the 1.17 Mb physical interval of *Ml3D232/Pm36* between the closest flanking markers *WGGBM2* and *WGGBM7*: *CYB561-Domon*, three *F-box* proteins (*F-box 1-3*), five *Serpin* proteins (*Serpin 1-5*), a tandem kinase protein with a single transmembrane domain in the C-terminus (*WTK7-TM*), and *TM9SF4* (Fig. 1d). Micro-collinearity analysis revealed highly conserved protein-coding genes in the *Pm36* genomic region among multiple wheat reference genomes[33–37], except for the WEW-durum introgression line 5BIL-29 with significant genomic structural variation (Supplementary Figs. 7 and 8). Compared to Zavitan, and all other available reference genomes, introgression line 5BIL-29 contains a ~900 kb insertion in the *Pm36* physical region harboring two segmental duplications containing 7 predicted genes (Fig. 1c, d; Supplementary Figs. 7, 8, and Supplementary Data 4). An about 70 kb fragment containing one *F-box* and two *Serpin* genes was likely duplicated twice. The first duplication was nearly intact with an additional 22 kb insertion resulting in 92 kb sequence. The second duplication contains the *F-box*, only one *Serpin*, the traces of 70 kb transposable elements, and a 300 kb sequence harboring the *WTK7-TM* (Supplementary Fig. 9).

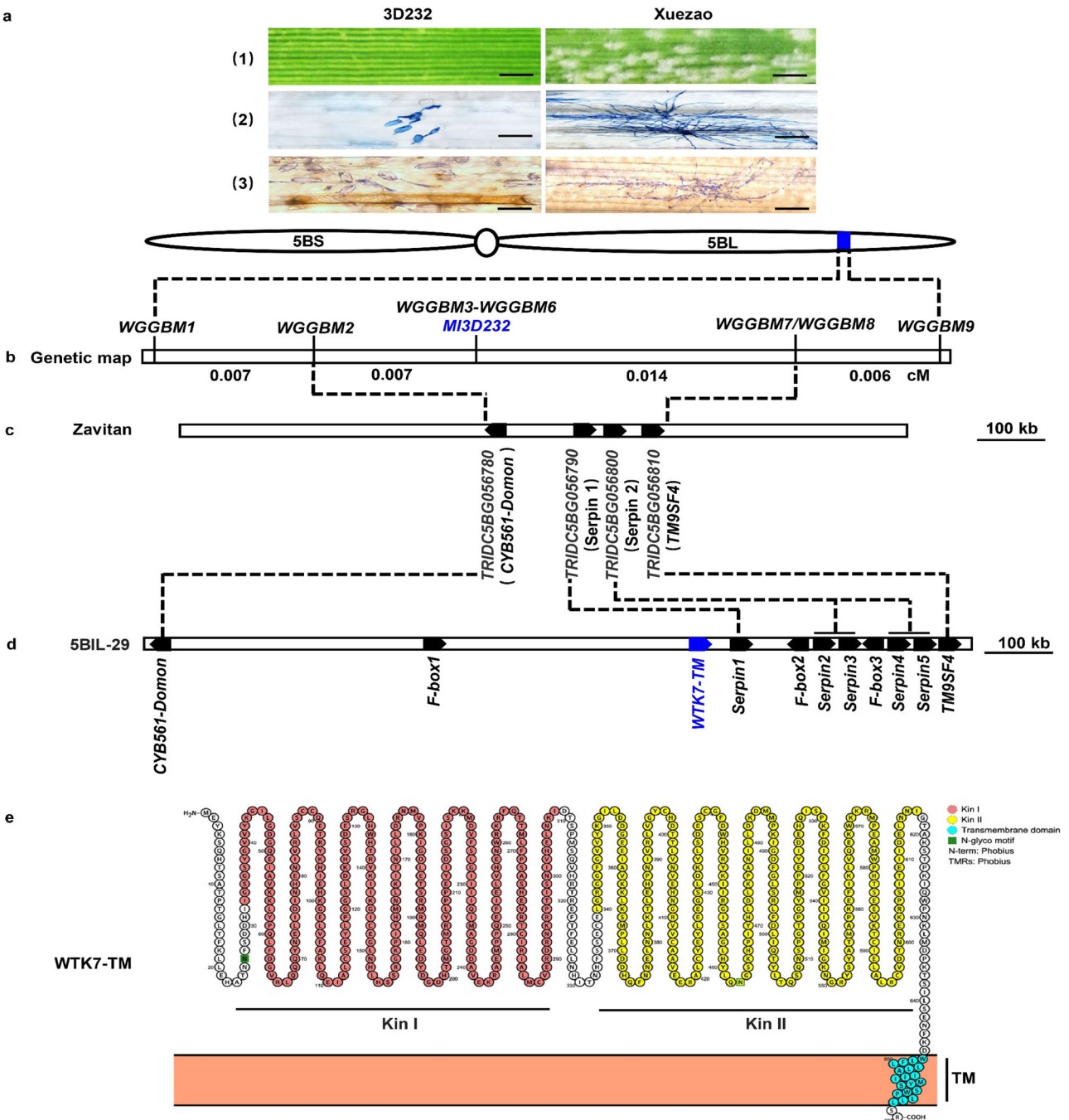

**Fig. 1 | Positional cloning of *Ml3D232*. a** Infection types of line 3D232 and Xuezao to *Bgt* isolate E09. Plants of 3D232 and Xuezao were inoculated with *Bgt* isolate E09 at the two-leaf stage. Plant leaves were detached and photographed at 10 day post-inoculation (dpi). Scale bar, 3 mm (1). Fungal structures of *Bgt* isolate E09 at 72 h post inoculation (hpi) were stained by Coomassie Brilliant Blue. Scale bar, 100 µm (2). DAB-Coomassie blue staining of infected leaves with *Bgt* isolate E09 from 3D232 and Xuezao at 48 hpi. Brown staining shows the accumulation of $H_2O_2$. Scale bar, 100 µm (3). Line 3D232 is immune to *Bgt* isolate E09, while Xuezao is highly susceptible to *Bgt* isolate E09. Three independent experiments were performed. **b** Genetic map of the region carrying *Ml3D232* on the long arm of Chinese Spring chromosome 5B. **c** Physical map covering the *Ml3D232* region on Zavitan reference genome (WEW_v1.0). **d** Physical map covering the *Ml3D232* region on tetraploid WEW-durum introgression line 5BIL-29 genome. **e** Protein prediction of wheat tandem kinase with single transmembrane (WTK7-TM). The kinase I (Kin I), kinase II (Kin II), and transmembrane (TM) domains of WTK7-TM are highlighted in orange, yellow, and cyan circles respectively. Source data are provided as a Source Data file.

RNA-seq analysis showed that only *CYB561-Domon*, *TM9SF4*, and *WTK7-TM* were expressed among the eleven annotated genes in the 1.17 Mb genomic interval in the leaves of 5BIL-29 seedlings after inoculation with *Bgt* isolate E09 (Supplementary Data 5). Given that *CYB561-Domon* and *TM9SF4* have been previously excluded by transgenic assay (Supplementary Fig. 6c), *WTK7-TM* was most likely the candidate gene for *Ml3D232/Pm36*.

## Cloning of *WTK7-TM*

We then cloned the 10,741 bp genomic DNA fragment of *WTK7-TM* including 3277 bp coding region, 3311 bp promoter region, and 4153 bp terminator sequence from lines 5BIL-29 and 3D232, and found no sequence variation in *WTK7-TM* between lines 5BIL-29 and 3D232. The full-length cDNAs of *WTK7-TM* were also obtained from line 3D232 by the 5′- and 3′-rapid amplification of

complementary DNA (cDNA) ends (RACE). Nine transcription isoforms of *WTK7-TM*, named as *TV1-TV9*, were identified, with isoform *TV1* being the most common among them (Supplementary Fig. 10). The *WTK7-TM* gene has 12 exons, which encodes a 667-residue tandem kinase protein including Kinase I (Kin I) and Kinase II (Kin II) domains, and a predicted single transmembrane domain in the C-terminus (Fig. 1e and Supplementary Fig. 11). Moreover, we found that *WTK7-TM* is constitutively expressed in the root, stem, and leaf tissues without *Bgt* infection (Supplementary Fig. 12a), but could be slightly upregulated after *Bgt* isolate E09 infection in lines 3D232 and 5BIL-29 (Supplementary Fig. 12b), prioritizing it for the further validation as a candidate gene of *Ml3D232/Pm36*.

### Validation of *WTK7-TM* by VIGS, EMS mutants, and transgenic assay

To examine whether *WTK7-TM* is responsible for the powdery mildew resistance of *Ml3D232/Pm36* in 3D232 and 5BIL-29, we performed virus-induced gene silencing (VIGS) experiment[38] with two constructs targeting the regions of Kin I and Kin II domains of WTK7-TM, respectively (Supplementary Fig. 13). The transcript levels of *WTK7-TM* in lines 3D232 and 5BIL-29 were substantially knocked down in both BSMV:*WTK7-TM*^Kin I and BSMV:*WTK7-TM*^Kin II infected plants compared to BSMV:00 infected plants (Fig. 2a and Supplementary Fig. 13). As a result, both lines 3D232 and 5BIL-29 lost their resistance to *Bgt* isolate E09 (Fig. 2b). The results indicated that *WTK7-TM* is necessary for resistance to *Bgt* in lines 3D232 and 5BIL-29.

To further validate the function of *WTK7-TM*, we used ethyl methane sulfonate (EMS) to mutagenize line 3D232 and identified 23 $M_3$ mutants highly susceptible to *Bgt* isolate E09 out of 3360 $M_3$ families (Fig. 2c and Supplementary Data 6). Further characterization with PCR and Sanger sequencing of the *WTK7-TM* gene including the promoter sequence, coding region, and 3' downstream sequence indicated that 17 mutants carried mutations within the CDS of *WTK7-TM*, including five premature stop codons, one frameshift alternative splicing variant, and 11 missense mutations (Fig. 2d and Supplementary Data 6). No sequence variations were detected at the *WTK7-TM* in the other 6 mutants, suggesting other factors may be involved in the regulation of *WTK7-TM* resistance. Seven mutants had mutations in the conserved Kin I domain and 8 mutants showed mutations in the Kin II domain, indicating that both Kin I and Kin II domains are essential for the function of WTK7-TM in conferring resistance to powdery mildew (Fig. 2d). Surprisingly, the loss of function mutant M16 carried a nonsense mutation in the transmembrane domain, generating a premature WTK protein without the transmembrane domain (Fig. 2c, d; Supplementary Fig. 14 and Supplementary Data 6). A point mutation was detected in mutant M17 in the splice acceptor site of intron 11, resulting in the frameshift alternative splicing manner with an extra 9 amino acids after the transmembrane domain (Fig. 2c, d; Supplementary Fig. 14 and Supplementary Data 6). This result demonstrated the importance of the transmembrane domain for the function of WTK7-TM. These independent mutations demonstrate that Kin I, Kin II, and TM domains of WTK7-TM are required for resistance against *Bgt*.

To further determine whether *WTK7-TM* is *Ml3D232/Pm36*, we cloned a 10,741 bp genomic fragment of *WTK7-TM* from 3D232, comprising the complete gene coding region and its regulatory elements, into the binary vector pCAMBIA-1300 (Fig. 3a) and transformed the susceptible wheat cultivar Fielder by *Agrobacterium*-mediated transformation. We obtained five independent $T_0$ transgene-positive plants by PCR analysis, which were self-pollinated and advanced to the $T_1$ generation. Evaluation of $T_1$ progeny of transgenic lines showed that individuals carrying the *WTK7-TM* transgene were resistant to *Bgt* isolate E09 (Fig. 3b, c). Taken together, these results show that *WTK7-TM* is the *Ml3D232/Pm36* gene.

### Subcellular localization, structure, and catalytic activity of WTK7-TM

Recent studies revealed that TKPs are key regulators of crop pathogen resistance, especially in wheat and barley[23]. However, all reported TKPs did not contain any membrane-targeting motif, except for WTK7-TM, suggesting a possible different resistance mechanism for this protein (Supplementary Fig. 15). Subcellular localization displayed WTK7-TM protein is localized in the plasma membrane in *N. benthamiana* (Supplementary Fig. 16). Phylogenetic analysis showed that both kinase domains of WTK7-TM are located in the same clade with Rpg1, WTK4, and WTK6-vWA that originate through kinase domain duplication, suggesting a possible duplication origin of WTK7-TM (Fig. 4). Based on Hidden Markov Model, both the two kinase domains of WTK7-TM belong to LRR_8B (cysteine-rich kinases) subfamily, like as Rpg1[24]. Further analysis by aligning WTK7-TM kinase domains with well-studied kinases indicated that Kin I and the Kin II domains of WTK7-TM are serine/threonine (S/T) non-arginine aspartate (non-RD) kinase with potentially catalytic activity, while two kinase domains are more likely pseudokinase due to the loss of some conserved residues (Supplementary Fig. 17). The three-dimensional (3D) model of WTK7-TM protein predicted by AlphaFold v2.0[39] revealed the Kin I and Kin II domains, and a single transmembrane domain in the C-terminus (Fig. 5a). Possessing two tandem kinase domains is the most distinct feature of TKPs, however, little is known on how these TKPs perform their kinase activities[23]. We, therefore, explored the WTK7-TM in vitro phosphorylation activity. The results showed that the full-length WTK7-TM (1–667 aa), Kin I (1–320 aa) or Kin II (321–646 aa) domain alone doesn't exhibit any auto-phosphorylation or catalytic activity, while the truncated version WTK7-TM^Kin I-Kin II (1–646 aa) composing of Kin I + Kin II without the transmembrane domain does demonstrate such activities (Fig. 5b, c and Supplementary Fig. 18).

### Geographic distribution of *WTK7-TM*

To evaluate the distribution of *WTK7-TM* in wild and cultivated wheat, we developed a functional marker *WTK7-TM-FM* for specific detection of *WTK7-TM* and screened wheat germplasm collected from geographically diverse sites in the world, including WEW (442 accessions), cultivated emmer (125 accessions), durum (230 accessions), and hexaploid wheat (1292 accessions) (Supplementary Data 7). *WTK7-TM* was detected only in 38 WEW accessions, which were resistant to *Bgt* isolates E09 and E20 (Supplementary Fig. 19 and Supplementary Data 8) and shared an identical genomic sequence and CDS of *WTK7-TM*. All of these WEW accessions originated from the southern (Israel, Lebanon, Jordan, and Syria) WEW populations and represent only 1.82% of the tested wheat germplasms (Supplementary Fig. 20). The absence of *WTK7-TM* from the other WEW populations, as well as domesticated emmer, durum, and common wheat suggests that *WTK7-TM* is likely an orphan gene with single ancient origin and will have a profound contribution to the current wheat gene pool.

## Discussion

Most characterized plant resistance proteins belong to the race-specific nucleotide-binding domain and leucine-rich repeat-containing (NLR) family that recognizes pathogen race-specific effectors[40]. Recently, a class of intracellular resistance proteins with tandem kinase domain architecture has emerged[23] and provides resistance against not only biotrophic fungal pathogens, *Puccinia graminis* f. sp. *tritici* (*Pgt*)[24,26,27], *P. striiformis* f. sp. *tritici* (*Pst*)[25], *P. triticina* f. sp. *tritici* (*Pt*)[28] and *B. graminis* f. sp. *tritici* (*Bgt*)[15,19,22], but also hemibiotrophic pathogen, *Pyricularia oryzae* (*Po*)[29]. Here, we isolated a broad-spectrum powdery mildew resistance gene *Pm36* (*WTK7-TM*) originated from wild emmer wheat by combining map-based cloning and PacBio SMRT long-read genome sequencing approach. *WTK7-TM* encodes a tandem kinase protein with a single transmembrane domain in its C-terminus. To date, all functionally

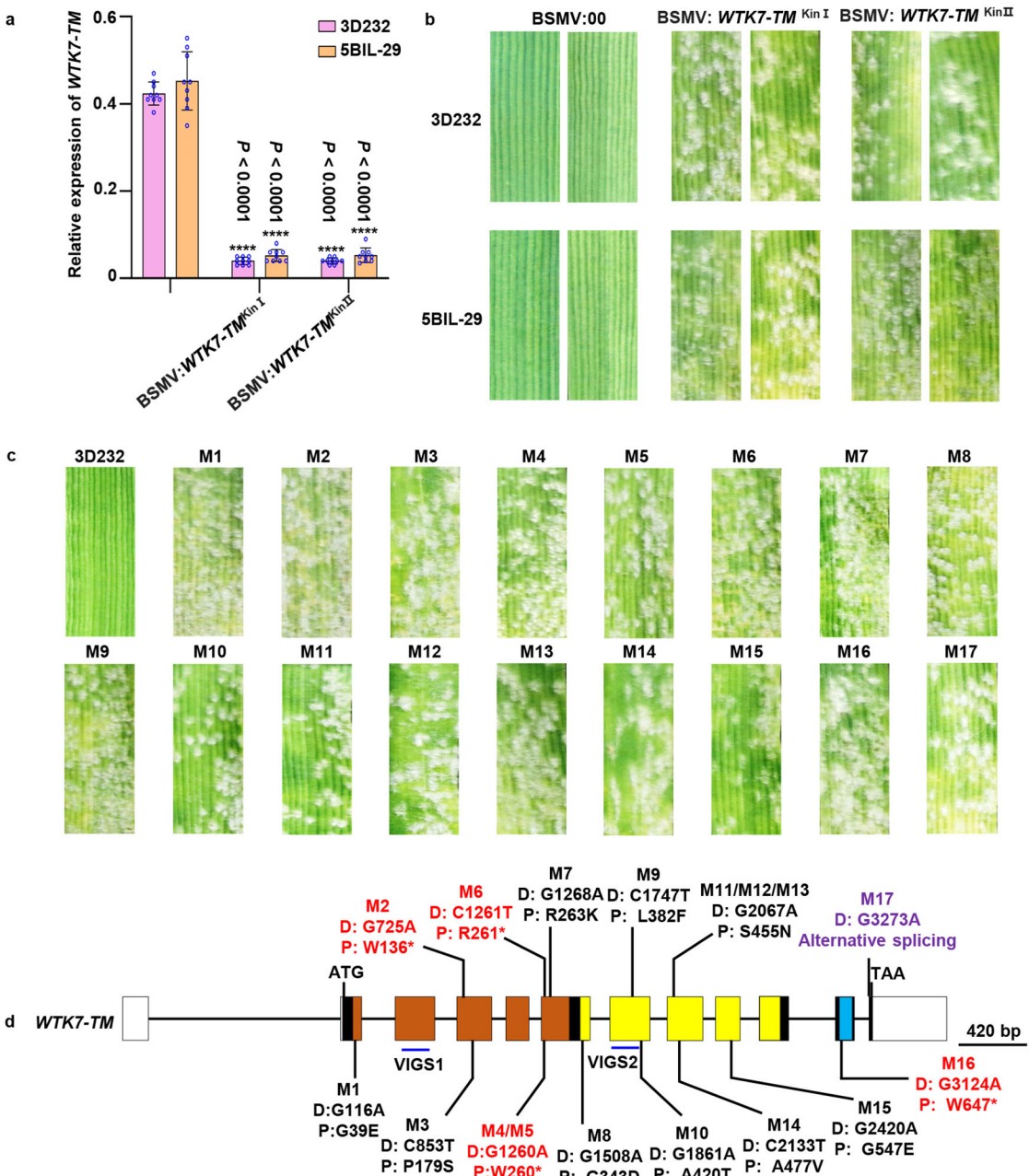

**Fig. 2 | Functional validation of *WTK7-TM* using VIGS and ethyl methane sulfonate (EMS) mutants assay. a** Relative expression levels of *WTK7-TM* in lines 3D232 and 5BIL-29 treated with BSMV:00 (control), BSMV:*WTK7-TM*Kin1 and BSMV:*WTK7-TM*KinII, respectively, were examined by qRT-PCR. Results represent the means ± SD from three biological replicates (three leaves used per biological replicate) and three technical replicates for each leave (*N* = 9, unpaired two-tailed *t*-test). "****" indicates a statistically significant difference (*P* < 0.0001). **b** Plants were inoculated with *Bgt* isolate E09 and representative leaves were photographed at 14 dpi. VIGS1 and VIGS2 showed the two target positions of *WTK7-TM* in BSMV-VIGS experiments. Three independent experiments were performed. **c** Line 3D232 and 17 susceptible mutants on *WTK7-TM* inoculated with *Bgt* isolate E09 and representative leaves were photographed at 14 dpi. Three independent experiments were performed. **d** Susceptible EMS mutants carrying single nonsense or missense mutations in the *WTK7-TM* gene. Black straight lines indicate introns, and rectangles indicate coding exons. Domains Kin I and Kin II, and the transmembrane region of WTK7-TM are highlighted in orange, yellow, and cyan rectangles, respectively. Short vertical lines indicate the mutated positions. The DNA sequence (D.) changes, and their predicted effects on the translated protein (P.) are indicated below the mutants. Mutation names in red, purple, and black indicate nonsense, splice site, and missense mutations, respectively. *P*-values and source data are provided as a Source Data file.

validated tandem kinase proteins in plants did not contain any membrane-targeting motif[23], except for WTK7-TM, suggesting a possible different resistance mechanism for this unique protein.

In plants, membrane-related proteins play essential roles in response to pathogens. Receptor-like proteins (RLPs) or receptor-like kinases (RLKs) containing a transmembrane (TM) domain and extracellular LRR domains, either a short domain or a kinase domain,

respectively, can recognize microbe - or plant-derived immunogenic molecular patterns from outside of the plant cell, resulting in the activation of immune responses[41]. *Arabidopsis thaliana* broad-spectrum mildew resistance gene *Rpw8* is specifically targeted to the extrahaustorial membrane where it promotes $H_2O_2$ accumulation and triggers a hypersensitive response to constrain the haustorium[42]. Barley leaf rust resistance gene *Rph3* encodes a unique

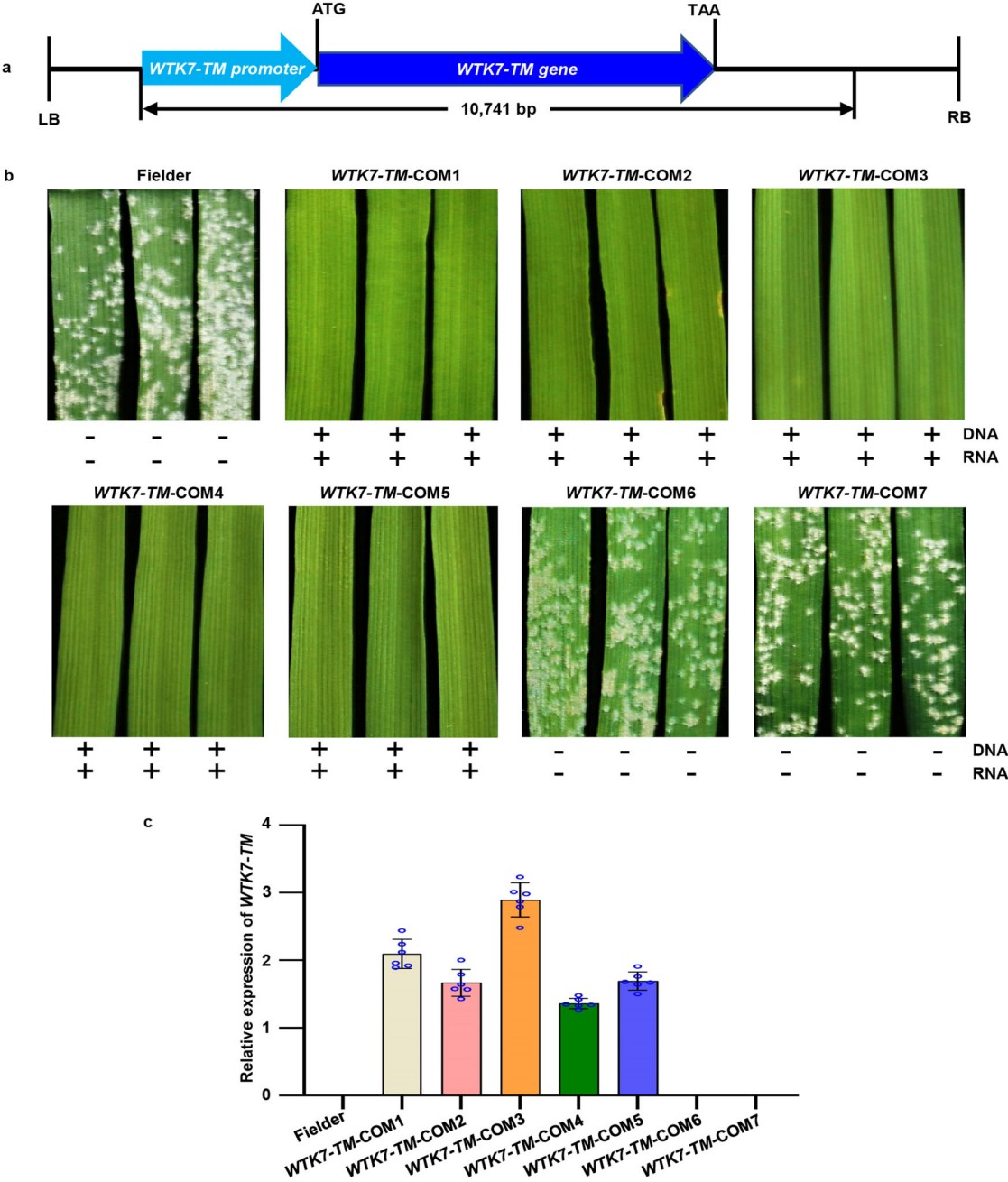

**Fig. 3 | Transgenic validation of *WTK7-TM*. a** Structure of *ProWTK7-TM:WTK7-TM* used for transgenic assay. The *ProWTK7-TM:WTK7-TM* construct contains the 3277 bp full-length *WTK7-TM* coding region, as well as 3311 bp upstream of the start codon and 4153 bp downstream of the stop codon. LB, left border; RB, right border. **b** The resistance performance of *ProWTK7-TM:WTK7-TM*–transformed transgenic plants. Transgenic T₁ wheat (about 15 plants per family) was challenged with *Bgt* isolate E09 at the two-leaf stage. Representative leaves were photographed at 12 dpi. PCR results as positive (+) or negative (-) for DNA amplification (upper) and RNA expression (lower) of the *WTK7-TM* gene. Three independent experiments were performed. **c** The relative expression levels of *WTK7-TM* in transgenic wheat. Leaves of seedling plants at the two-leaf stage were collected. *TaActin* was used as an endogenous control. Expression values are means ± SD from three biological replicates (three leaves used per biological replicate) three biological replicates (three leaves used per biological replicate) and three technical replicates for each leave (*N* = 9). There was no expression in non-transgenic plants. Error bars are standard errors of the mean. Source data are provided as a Source Data file.

predicted transmembrane resistance protein and is expressed only after a challenge by rust isolates[43]. Several barley and wheat resistance genes also encode putative transmembrane proteins or integrated with additional functional domains. Broad-spectrum powdery mildew resistance gene *mlo* encoding a seven-transmembrane (TM) protein acts as negative regulators of powdery mildew resistance[44]. The wheat durable, multi-pathogen resistance genes *Lr34*[16] and *Lr67*[18] encode multiple transmembrane proteins which belong to ATP-binding cassette (ABC) transporter

and hexose transporter, respectively. *Pm4* encodes a putative chimeric protein of a serine/threonine kinase and C2 domains and transmembrane regions, which localize to the endoplasmatic reticulum[7]. Membrane-localized protein *Lr14a* containing twelve ankyrin (ANK) repeats and structural similarities to Ca²⁺-permeable non-selective cation channels confers race-specific leaf rust disease resistance[45]. The *Stb16q* gene encodes a plasma membrane cysteine-rich receptor-like kinase with the DUF26 domain, a transmembrane domain, and serine/threonine protein kinase domain[46]. In

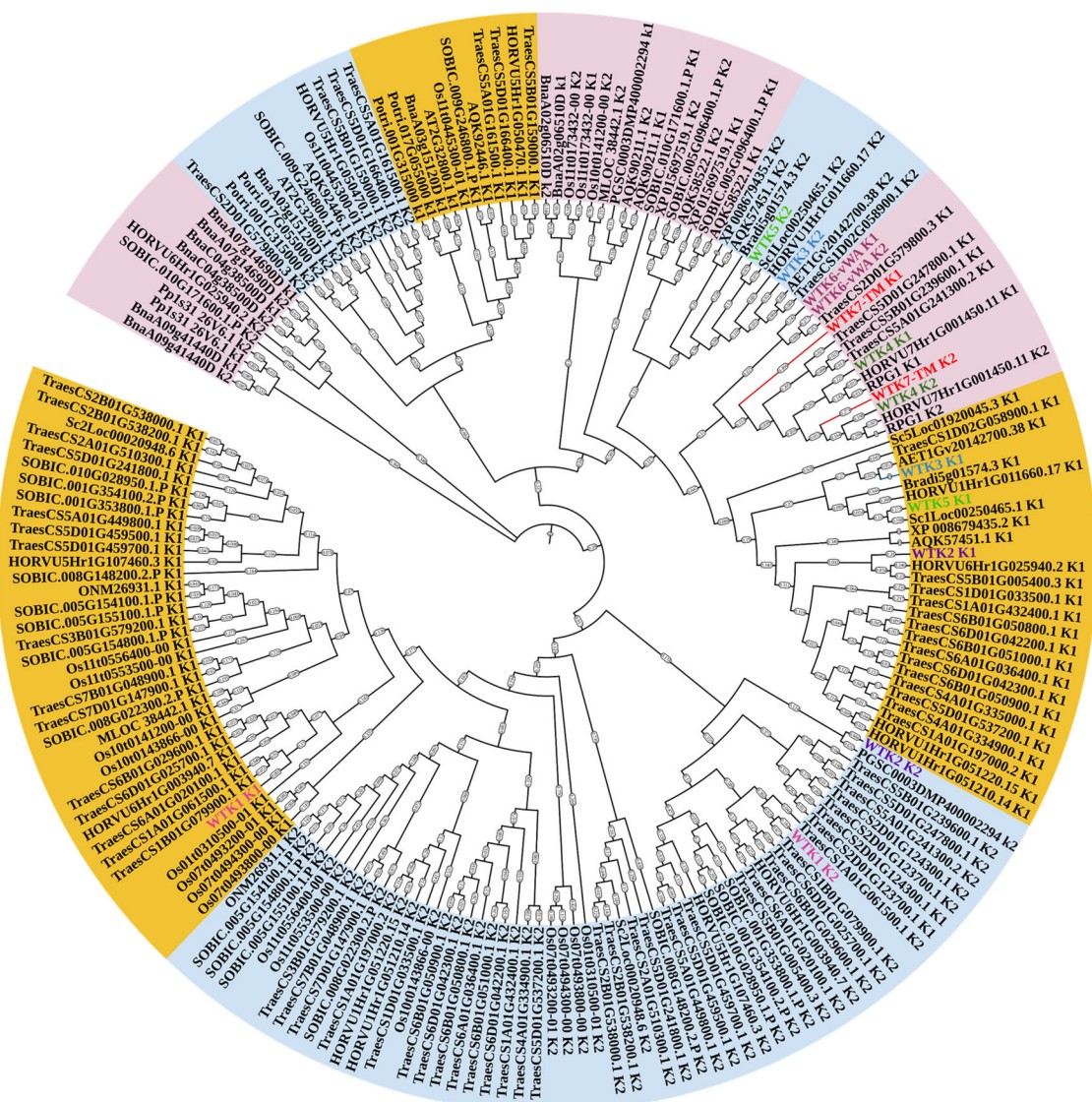

**Fig. 4 | Phylogenetic analysis of predicted tandem kinase in plants.** Phylogenetic tree was built using 196 putative kinase or pseudokinase domains with the Neighbor-joining method. ClustalW in the MEGA was used for multiple alignment using the default setting and phylogenetic construction. Phylogenetic tree was polished by iTOL.

the present study, WTK7-TM is a membrane-related protein with tandem kinase domains. Genetic evidence show that EMS mutant M16 carried an early stop codon mutation before the transmembrane region, resulting in a truncated WTK7-TM protein without the transmembrane domain. Another EMS mutant M17 has an extended C-terminus in the predicted transmembrane domain region due to the frameshift. Those results revealed the essential role of the transmembrane domain for WTK7-TM powdery mildew resistance (Supplementary Fig. 14). Protein localization analysis showed that WTK7-TM protein is localized in not only cytomembrane but also the nucleus and cytoplasm in the wheat leaf mesophyll protoplasts (Supplementary Fig. 16). It will be interesting to investigate trafficking of WTK7-TM in pathogen-plant interaction.

The most typical protein architecture of TKPs includes two kinase-like domains (Kin I and Kin II)[23]. It can be generated by different molecular mechanisms, including gene duplication, and fusion according to high or low protein sequence similarity and kinase families. WTK1[25], WTK2[26], WTK3[15] and WTK5[27] classified with fusion origin, while WTK4[22], WTK6-vWA[28], and RPG1[24] with duplication origin. Phylogenetic analysis of individual kinase domains indicated that the Kin I and Kin II domains of WTK7-TM are quite close to Kin I and Kin II

of Rpg1, WTK4, and WTK6-vWA with relatively high amino acid sequence identity between the Kin I and Kin II domains, implying duplication origin (Fig. 4). Based on the conservation of the key amino acid residues for kinase domain, WTK1[25], WTK3[15], WTK5[27], and WTK6-vWA[28] can be classified as kinase-pseudokinase structure, WTK4[22] and Rpg1[24] as pseudokinase-kinase structure, while WTK2[26] as kinase-kinase structure. Pseudokinase of TKPs may lose the function of phosphorylation and gain function as an effector decoy in plant–pathogen interactions[23]. In this study, though comparing with reported plant kinase proteins, both Kin I and Kin II of WTK7-TM are more likely pseudokinase due to the loss of some conserved residues. The EMS-induced susceptible mutants had mutations in the conserved Kin I and Kin II domains of WTK7-TM, implying that both Kin I and Kin II domains of WTK7-TM are critical for their powdery mildew resistance function. However, the phosphorylation experiment showed that the full-length WTK7-TM, Kin I or Kin II domain alone doesn't exhibit any auto-phosphorylation or catalytic activity, while the truncated version, MBP-WTK7-TM$^{\text{Kin I-Kin II}}$ does demonstrate such activities in vitro (Fig. 5b, c and Supplementary Fig. 18). We speculate that the full-length WTK7-TM phosphorylation is required for recognition of *Blumeria graminis* f. sp. *tritici* (*Bgt*) in vivo which is similar with reported RPG1

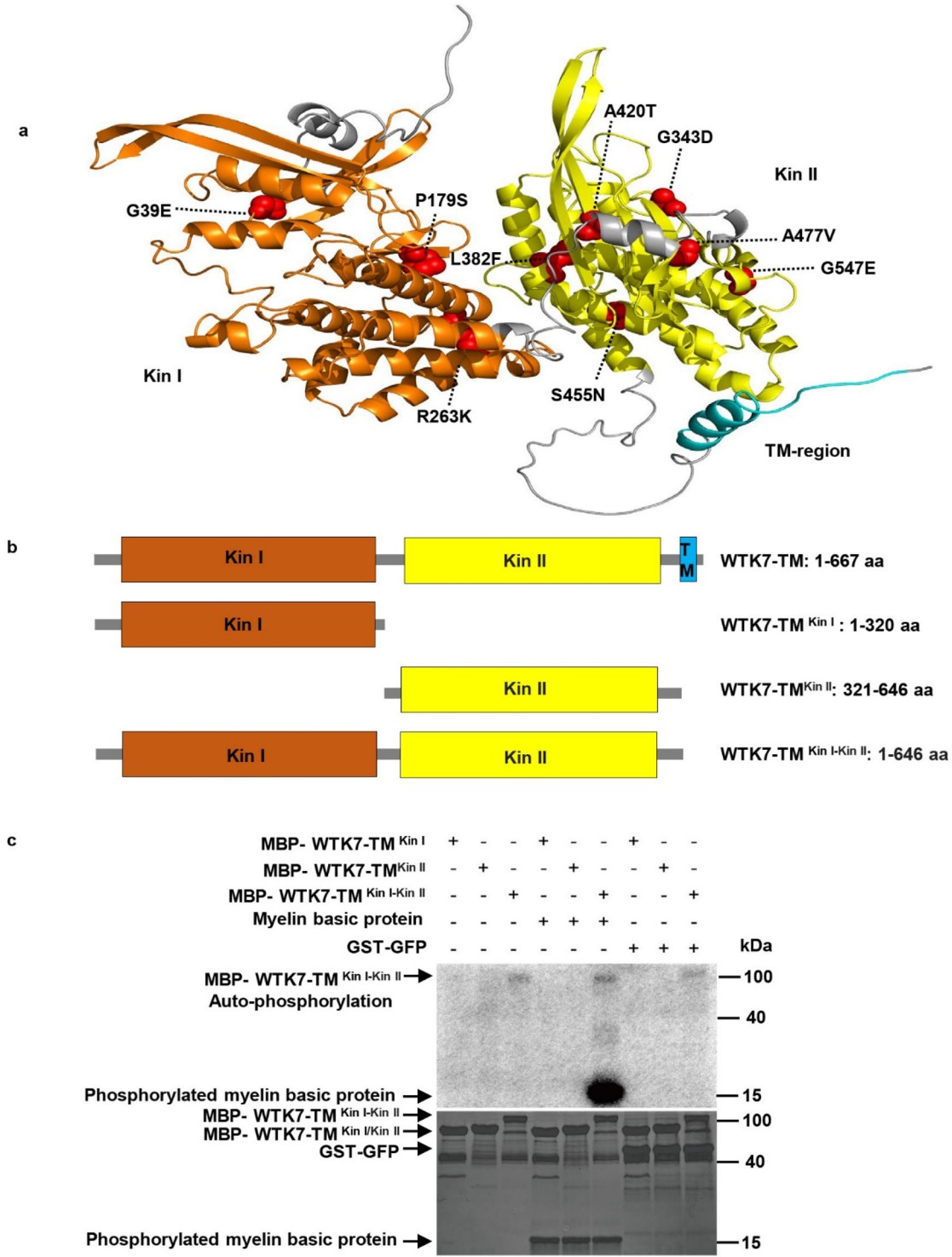

**Fig. 5 | Protein structure prediction, and kinase activity of WTK7-TM. a** WTK7-TM protein structure was predicted on AlphaFold v2.0. Domains Kin I and Kin II, and the transmembrane region of WTK7-TM are highlighted in orange, yellow, and cyan, respectively. Mutations on domains Kin I and Kin II of WTK7-TM are indicated by red spheres. **b** Schematic diagram of the full-length protein WTK7-TM (1–667 aa), Kin I (1–320 aa), Kin II (321–646 aa) and WTK7-TM^Kin I-Kin II consisting of Kin I + Kin II without transmembrane of WTK7-TM (1–646 aa).

**c** Kinase activity analysis of truncated WTK7-TM in vitro. Neither Kin I nor Kin II domains have kinase activity while WTK7-TM^Kin I-Kin II (1–646 aa) has auto-phosphorylation and can phosphorylate myeline basic protein. Autoradiograph and coomassie brilliant blue staining are shown in the top and bottom panels, respectively. GST-GFP was used as a substrate for negative control. Three independent experiments were performed. Source data are provided as a Source Data file.

phosphorylation depending on the recognition of *Puccinia graminis* f. sp. *tritici*[24].

Understanding the genetic variation that underpins phenotypic variation is necessary for cloning of agronomically important genes in crop plants. Genome structural variations (SVs), which include large deletions, insertions, duplications, and chromosomal rearrangements, are common in crop genomes[47]. Although multiple wheat genome sequences and abundant variations are already available[33–37], cloning of

agronomically important genes, including disease resistance genes, remains a challenging task at a particular locus. Physical mapping and sequence comparison indicated an approximate 136 kb fragment deletion was present in the corresponding *Pm41* genomic region in Chinese Spring chromosome 3BL[17]. Maize *ZmWUS1* locus exists a large tandem segmental duplication event that created a unique 119 bp insertion in the proximal promoter region of the duplicate *ZmWUS1-B* copy[48]. In our study, we sequenced a tetraploid WEW-durum wheat introgression line 5BIL-29 carrying *Pm36* using PacBio SMRT long-read sequencing approach. After sequence assembly, we found that *Pm36* locus underwent a ~ 900 kb insertion harboring an additional 7 annotated genes including *WTK7-TM* with two segmental duplications compared to Zavitan and other available wheat reference genomes. The finding also demonstrates structural variations are major obstacles to gene identification in complex genomes, such as wheat and wild relatives, based on the limited availability of reference and pan-genome sequences. Recently, long-read sequencing technologies have also been used to clone wheat yellow rust resistance gene *Yr27* [49] and powdery mildew resistance gene *Pm69*[21], indicating a powerful tactic for facilitating disease resistance gene cloning in wheat.

The crop wild relatives serve as an important untapped reservoir of genetic diversity that can be used to improve modern breeding. The process of wheat domestication and hexaploidization has significantly reduced genetic diversity relative to the wild ancestor across the genome[50]. *WTK7-TM* was found only in the southern WEW populations and seems to be absent in other WEW populations, as well as domesticated emmer, durum, and common wheat, indicating a relatively ancient single origin. The geographical distribution of *WTK7-TM* is similar with *Pm41*[17], *Yr15*[25], and *Yr36*[51] appearing in the Mount Carmel to Anti-Lebanon mountain ridge, manifesting host–parasite coevolution in WEW natural habitats. Although more than a hundred *Pm* genes have been reported, only a few can provide resistance to most or all *Bgt* isolates, such as *Pm12* from *Aegilops speltoides*[10], *Pm21*[13,14] from *Dasypyrum villosum* L., and *Pm24*[15] from common wheat. *Pm38/Yr18/Lr34/Sr57*[16] and *Pm46/Yr46/Lr67/Sr55*[18] provide durable resistance to powdery mildew, leaf rust, stem rust, and stripe rust. *Pm21* confers immunity to all tested *Bgt* isolates and has been widely utilized in wheat breeding in China since 1995[52]. The powdery mildew resistance gene *Pm36* was transferred to common wheat line 3D232 from wild emmer accession I222 in 2003, and it is still one of the most effective resistance genes in China. Wheat line 3D232 showed broad-spectrum resistance against genetically diverse *Bgt* isolates (Supplementary Fig. 1 and Supplementary Data 1) without a growth penalty. Yet, *Bgt*-resistant wheat cultivars carrying a single race-specific *Pm* gene are prone to lose their resistance due to strong pathogen selective pressure. Therefore, combining multiple and diverse resistance genes is a sustainable approach to disease management in wheat breeding. Our discovery provides a promising gene for designing future wheat with broad-spectrum resistance to powdery mildew.

## Methods
### Plant materials
Wild emmer accession I222 was kindly provided by Dr. Z. K. Gerechter-Amitai, Agricultural Research Organization, The Volcani Center, Israel. Common wheat line 3D232 was developed by Professor Tsomin Yang, China Agricultural University, China, by backcrossing a susceptible common wheat line 87-1 with wild emmer accession I222 (pedigree: 87-1*6/I222, $BC_5F_6$)[31]. A single dominant gene *Ml3D232* conferring powdery mildew resistance has been mapped to the chromosome arm 5BL (bin 5BL 0.59–0.76) using a molecular and deletion mapping strategy[31]. A backcross inbred line 5BIL-29 carrying powdery mildew resistance gene *Pm36* was previously developed by crossing wild emmer wheat accession MG29896 with durum wheat cv. Latino[30] and kindly provided by Dr. Antonio Blanco, Department of Soil, Plant and Food Sciences

(DiSSPA), Genetics and Plant Breeding Section, University of Bari Aldo Moro, Italy. Wheat collections for geographical distribution analysis of *Ml3D232/Pm36* included 442 wild emmer wheat (*T. turgidum* ssp. *dicoccoides*), 125 cultivated emmer wheat (*T. turgidum* subsp. *dicoccum*), 230 durum (*T. turgidum* subsp. *durum*), 262 entries of Chinese wheat mini-core collection (MCC)[53], 636 and 394 common wheat accessions from China and the other 34 countries, respectively[8]. Most of these accessions were obtained from the Wheat Genetic and Genomic Resources Center, Kansas State University, USA, The USDA-ARS National Small Grains Collection, USA, and the University of Haifa, Israel. The 262 entries of Chinese wheat mini-core collection (MCC) were available in the National Key Facility for Crop Gene Resources and Genetic Improvement, Chinese Academy of Agriculture Sciences. The 125 cultivated emmer wheat (*T. turgidum* subsp. *dicoccum*), 230 durum (*T. turgidum* subsp. *durum*) have the official Plant ID, such as PI ×××, CItr ××× as shown in Supplementary Data 7. Plants were grown in a greenhouse with 16 h light/8 h dark (24/18 °C, 70% relative humidity).

### Powdery mildew assays
One hundred and four *Bgt* isolates, collected from geographically different sources in China were used to determine reactions of wheat entries to powdery mildew (Supplementary Fig. 1 and Supplementary Data 1). *Bgt* isolates E09 was used to inoculate the parental lines, genetic segregating populations, ethyl methane sulfonate (EMS)-induced mutants, transgenic plants, and wheat collections and also used for BSMV-VIGS and gene expression analysis. Evaluation of the seedling reactions to *Bgt* isolates was carried out in a greenhouse at the Institute of Genetics and Developmental Biology, Chinese Academy of Sciences, Beijing, China. All tested plants were inoculated at the two-leaf stage under controlled glasshouse conditions[15]. Powdery mildew infection types (ITs) on the primary leaf of each plant were rated using a 0–4 scale at 7–10 days post inoculation (dpi) when the susceptible control Xuezao displayed severe symptoms and re-examined 2 days later. Based on IT scores, tested plants were classified as highly resistant (IT = 0, 0;, and 1), moderately resistant (IT = 2), moderately susceptible (IT = 3), and highly susceptible (IT = 4). At the jointing stage, lines 3D232 and Xuezao were inoculated with *Bgt* isolate E09 under field conditions at the Gaoyi Experimental Station at Shijiazhuang in Hebei province. Powdery mildew reactions were recorded on a 0–4 scale at the grain-filling stage when susceptible Xuezao control plants were heavily diseased[15].

To visualize the fungal structures and detect the accumulation of $H_2O_2$, detached leaves from seedlings of 3D232 and Xuezao at 72 h post-inoculation (hpi) inoculated with *Bgt* isolate E09 were incubated in a 3,3'-diaminobenzidine (DAB) solution (1 mg/mL, PH 5.8) for 12 h, bleached in absolute ethanol and then stained with a 0.6% (w/v) Coomassie Brilliant Blue solution for 10 s and then washed with water[54]. Bright-field images were captured using a positive fluorescence microscope, Olympus BX53F (Olympus, Tokyo, Japan).

### Fine mapping of *Ml3D232/Pm36*
To expedite the fine mapping of *Ml3D232*, we developed a large $F_2$ population with 15,893 plants derived from the cross between the susceptible common wheat line Xuezao and the resistant common wheat line 3D232. Total genomic DNA was extracted from leaves using the CTAB procedure[55]. Genetic markers linked to *Ml3D232/Pm36* including SSR and dCAPs markers (Supplementary Data 2) were developed using the Chinese Spring (RefSeq v1.0)[35] and the WEW accession Zavitan reference genome (WEW_v1.0)[33] sequences for constructing a high-resolution genetic linkage map. The $F_2$ population from the Xuezao × 3D232 cross was first genotyped with SSR markers *XBD37670* and *XBD37760*, which flank *Ml3D232*[32], and individuals with recombination events between these markers were then genotyped with all the other markers (Supplementary Data 2) and subjected to

powdery mildew assay for the $F_3$ progenies. The linkage between genetic markers and *Ml3D232* was established using JoinMap4.0 (http://www.kyazma.nl/index.php/mc.Join-Map/sc.General/) with a LOD threshold of 3.0. Genetic map distances were calculated by means of the Kosambi mapping function[56].

## PacBio HiFi and Hi-C sequencing

High molecular weight DNA was isolated from 20-day-old fresh leaf tissue of 5BIL-29 carrying *Pm36* following the large-scale nucleus extraction protocols[57]. DNA purity was assessed based on the ratio of absorbance at 260 nm and 280 nm (A260/A280) using a NanoDrop NP-1000 spectrophotometer (NanoDrop Technologies, Wilmington, USA). DNA yield was assessed using the Qubit dsDNA HS Assay kit (Thermo Fisher Scientific, Waltham, USA), and DNA size was validated by pulsed-field gel electrophoresis (PFGE) of Pippin pulse. DNA was sheared to the appropriate size range (15−20 kb) using a Covaris g-TUBE for the construction of PacBio HiFi sequencing libraries[58], followed by bead purification with PB Beads (PacBio). HiFi sequencing libraries were prepared using SMRTbell Express Template Prep Kit 2.0 and followed by immediate treatment with the Enzyme Clean Up Kit. The libraries were further size-selected electrophoretically using BluePippin Systems from SAGE Science. Sequencing was performed on the Pacific Biosciences Sequel II system using HiFi sequencing protocols. Libraries were sequenced on six PacBio SMART cells (Supplementary Data 3).

For chromosome conformation capture (Hi-C) sequencing, isolated DNA from 20-day-old 5BIL-29 fresh leaf tissue was cross-linked and fixed. The fixed samples were sent to Wuhan Grandomics Biosciences Co., Ltd, China, for Hi-C library construction and sequencing. The quality of the libraries was assessed with Agilent 2100 Bioanalyzer (Agilent), and subjected to 2 × 150 bp paired-end high-throughput sequencing by Illumina NovaSeq/MGI. Finally, 1220.8 Gb of effective data were obtained (Supplementary Data 3).

## Genome assembly and gene annotation

The cumulative length of subreads from each of the six PacBio SMART cells ranged from 455.6 to 530.3 Gb, resulting in a combined length of HiFi reads ranging from 29.61 to 35.26 Gb, total 202.14 Gb. The N50 for HiFi reads ranged from 15.9 to 23.2 kb, with the max subreads length 65 kb. All the HiFi and Hi-C reads were assembled using the default parameters of Hifiasm 0.16.1-r375[59], and the gfatools (https://github.com/lh3/gfatools) was used to convert sequence graphs in the GFA to FASTA format. We captured a 7.1 Mb contig ptg000422l using flanking and co-segregating markers (*WGGBM2* to *WGGBM7*) of *Ml3D232*, which can entirely cover the *Ml3D232/Pm36* physical mapping interval (1.17 Mb). This 1.17 Mb genomic sequence was annotated using Softberry FGENESH[60] and NCBI BLASTP against the nr Database to identify homologous proteins. Repetitive sequences and transposable elements (TEs) were identified through BLASTN against the *Triticeae* repetitive elements (TREP) (https://trep-db.uzh.ch/). The long terminal repeat sequences of LTR retrotransposable elements were delineated using Dotter analysis and manually checked. The evolutionary distance of the two LTR sequences was estimated by MEGA11 (https://www.megasoftware.net/) using a substitution rate of $1.3 \times 10^{-8}$ mutations per site per year.

## RNA-seq, quantitative real-time PCR, and RT-PCR

Total RNA samples were extracted from two-leaf stage seedling leaves of 3D232 and 5BIL-29 at 0, 6, 12, 24, 36, 48, and 72 hpi with *Bgt* isolate E09 using the TRIzol reagent (Tiangen, Beijing, China). The first-strand cDNA from total RNA was synthesized using PrimeScript™ RT reagent Kit with gDNA Eraser (Perfect Real Time) (TaKaRa, Kyoto, Japan), and for quantitative real-time PCR (qRT-PCR) was performed with SYBR green kit (TaKaRa, Kyoto, Japan) in a Roche 480 light cycler (Roche, Colorado Springs, CO, USA). *TaActin*

was used as the internal control, and the $2^{-\Delta\Delta Ct}$ method was used to calculate relative gene expression[61]. To investigate whether *WTK7-TM* is expressed in different wheat tissues, we conducted RT-PCR analysis in the leaf, stem, and root of 3D232 at the two-leaf stage. *TaActin* was used as the endogenous control. Three biological replicates (three leaves or roots or stems used per biological replicate) from three independent experiments and three technical replicates for each sample were performed.

For RNA-seq, the RNA samples from two-leaf stage seedling leaves of 3D232 or 5BIL-29 inoculated with *Bgt* isolate E09 at 24hpi were selected to construct cDNA libraries and subjected to high-throughput paired-end sequencing by Illumina HiSeq2000 at Beijing Novogene Bioinformatics Technology Co. Ltd, Beijing, China. Clean reads were aligned to the reference genome of Chinese Spring or contig ptg000422l by Hisat2[62], and SNPs were called by GATK haplotypeCaller[63] following the GATK best practice for RNA-Seq.

## 3′- and 5′-RACE

Total RNA was isolated from two-leaf stage seedling leaves of 3D232 inoculated with *Bgt* isolate E09 at 24 hpi, using the TRIzol reagent (Tiangen, Beijing, China). Reverse transcription was performed using a PrimeScript RT reagent Kit with a gDNA Eraser (TaKaRa, Kyoto, Japan). We perform a rapid amplification of cDNA ends (RACE) to determine the transcriptional start (5′-RACE) and end (3′-RACE) of the *WTK7-TM* gene. The 3′- and 5′-UTR sequences of *WTK7-TM* were identified using the SMARTer RACE cDNA Amplification Kit (cat. no. 634923; Clontech, Mountain View, CA, USA). One hundred colonies were sequenced using the Sanger sequencing.

## Virus-induced gene silencing

We created gene-silencing constructs by cloning fragments of *WTK7-TM* into the BSMV RNA γ-derived binary vector pCa-γbLIC[38]. Two fragments of *WTK7-TM*, one from the Kin I domain and the other from the Kin II domain were amplified from recombinant vector pEASY:*WTK7-TM TV1* carrying *TV1* transcription isoform of *WTK7-TM* and inserted separately into pCa-γbLIC to generate independent recombinant constructs, BSMV:*WTK7-TM*^Kin I, and BSMV:*WTK7-TM*^Kin II. Firstly, equal amounts of *Agrobacterium tumefaciens* strain GV3101 harboring pCaBS-α, pCaBS-β and pCa-γbLIC (or BSMV:*WTK7-TM*^Kin I, and BSMV:*WTK7-TM*^Kin II) were mixed and infiltrated into *N. benthamiana* leaves at four to eight-leaf stage with a 1-mL needleless syringe. At 10 dpi, the infiltrated *N. benthamiana* leaves were ground in 20 mM Na-phosphate buffer (pH 7.2) containing 1% celite[38], and the sap was mechanically inoculated onto 3D232 and 5BIL-29 leaves at the two-leaf stage. Three weeks after inoculation with the *N. benthamiana* leaf sap, wheat developing visible BSMV symptoms (stripe mosaic on the 3th leaf with a mild mosaic on the 4th leaf) was challenged with *Bgt* isolate E09. Powdery mildew reactions of each plant were rated on a 0−4 IT scale[15]. The expression levels of *WTK7-TM* were also determined by qRT-PCR.

## Ethyl methane sulfonate mutagenesis and screening

Ethyl methane sulfonate (EMS) mutagenesis of 3D232 was performed with a concentration of 0.6% EMS (M0880, Sigma-Aldrich, Shanghai, China). Seeds were soaked in water for 12 h, incubated for 16 h with shaking in EMS solution and washed with running water for 4 h at room temperatures[64]. Mutagenized seeds were grown in the field at Gaoyi Experimental Station, Shijiazhuang, Hebei province, and 3360 $M_1$ plants were harvested. $M_2$ families (15−25 seeds per family) were screened for their response to *Bgt* isolate E09 under glasshouse conditions. Susceptible $M_2$ plants were transplanted and grown in pots and the $M_3$ progenies were challenged again with *Bgt* isolate E09 to confirm their susceptibility, as well as for authenticity of genotype using an array of SSR markers (Supplementary Data 2). Finally, we obtained 23 independently susceptible $M_3$ mutants of

3D232. The full-length genomic sequence of the *WTK7-TM* including the promoter, coding region, and terminator sequence from each of the 23 mutants was obtained using the primers, WTK7-60, WTK7-61, and WTK7-62 listed in Supplementary Data 2 and compared with the wild-type 3D232 using DNAMAN 8 software (Lynnon Biosoft, San Ramon, CA, USA).

### Wheat transformation

The coding sequence (CDS) of the *CYBS61-Domon* and *TM9SF4* genes were amplified from 3D232 using the primers listed in Supplementary Data 2 and inserted downstream of the maize (*Zea mays* L.) *Ubiquitin* (*Ubi*) promoter in a pTCK303 vector digested by *Kpn* I and *Spe* I to generate the overexpression vector *ProUbi*: C*YBS61-Domon* and *ProU-bi:TM9SF4*, respectively. A 10,741 bp genomic DNA fragment that contains the 3277 bp full-length *WTK7-TM* coding region, as well as 3311 bp upstream of the start codon and 4,153 bp downstream of the stop codon, was cloned into the vector pCAMBIA1300 to construct *ProWTK7-TM: WTK7-TM* for transformation. These resulting plasmids were transformed into the *Agrobacterium tumefaciens* strain EHA105 and delivered into the soft white spring wheat cultivar Fielder by the *A. tumefaciens*-mediated transformation method, respectively[65]. Fifteen individuals of T₁ transgenic families were inoculated with *Bgt* isolate E09. The transgenic plants were identified by PCR amplification and sequencing. The gene expression was measured with qRT-PCR as mentioned above. *TaActin* was used as an endogenous control. Expression values are means ± SD from three biological replicates (three leaves used per biological replicate) and three technical replicates for each leaf. All primers used for the construction preparation, the transgenic plant identification, and gene expression are listed in Supplementary Data 2.

### Subcellular localization assay

For subcellular localization, the coding sequence of *WTK7-TM* was inserted into the binary vector pCAMBIA1300-GFP driven by the 35 S promoter to construct the recombinant plasmid 35 S:: WTK7-TM-GFP. *Arabidopsis* NAA60 was used as a plasma membrane marker protein described by Linster et al.[66]. Fusion proteins 35 S::WTK7-TM-GFP + 35 S::NAA60-mCherry or 35 S::GFP + 35 S::NAA60-mCherry were expressed in *N. benthamiana* leaves through agroinfiltration. For plasmolysis experiments, leaf samples were incubated for 10–15 min in 4% NaCl before imaging[45]. *N. benthamiana* leaves were imaged 2dpi using a confocal laser scanning microscope (Carl Zeiss, LSM880).

### Protein extraction and analysis

*N. benthamiana* leaves were frozen in liquid nitrogen immediately after harvest. Subsequently, the leaves were finely ground into powder and transferred into 1.5 mL centrifuge tubes. Total proteins were extracted using lysis buffer (10 mM Tris-HCl (pH 7.6), 150 mM NaCl, 2 mM EDTA, 0.5% NP-40, 10 mM DTT, 1×complete protease inhibitor cocktail (Roche) and 1 mM phenylmethylsulfonyl fluoride (PMSF)). After vortexing for 20 min at 4 °C, the samples were centrifuged at $14,000 \times g$ for 10 min at 4 °C. Then transferred the supernatant to a new 1.5 mL Eppendorf tube. The protein samples were boiled with 1×SDS loading buffer and analyzed by immunoblotting. The proteins were detected by SDS-PAGE using Anti-GFP pAb-HRP-DirecT (MBL, 1:3000 dilution) antibodies.

### In vitro phosphorylation

For the in vitro phosphorylation assay, the indicated lengths of WTK7-TM CDS were cloned into the vector pMAL-c5x to obtain *MBP-WTK7-TM*[Kin I], *MBP-WTK7-TM*[Kin II], *MBP-WTK7-TM*[Kin I-Kin II] and *MBP-WTK7-TM-FL* constructs. Recombinant proteins were induced in DE3 (Rosetta) strain with 0.4 mM isopropyl-β-D-thiogalactopyranoside (IPTG) for 12 h at

16 °C. 100 mL cell culture was collected at 4 °C by centrifugation and the pellet was resuspended in 20 mL PBS buffer (200 mM NaCl, 10 mM Na₂HPO₄, 2.7 mM KCl, 1.8 mM KH₂PO₄, pH 7.4) with 1 mM PMSF, 5 mM DTT and 0.05% Triton-X 100. The suspension was sonicated for 8 min and centrifugated at $8000 \times g$ for 25 min. Then the supernatant was incubated with amylose resin for 2–3 h at 4 °C and MBP-tagged proteins were eluted with 10 mM maltose in PBS buffer. The purified proteins were separated by 10% SDS-PAGE and stained with coomassie brilliant blue to examine the purity. 5 μg indicated MBP-tagged proteins were incubated with 2 μg myelin basic protein in kinase reaction buffer (50 mM Hepes (pH 7.5), 10 mM MgCl₂, 1 mM DTT, 0.1 mM ATP, 1 μCi [γ-$^{32}$P]-ATP) for 20 μL volume. After being kept in 30 °C for 30 min, 5×SDS loading buffer was added into the mixture to stop the reaction and the proteins were separated by 12% sodium dodecyl sulfate polyacrylamide gel electrophoresis (SDS-PAGE). The radioactivity was detected with the Typhoon FLA 9500 (GE Healthcare Life Sciences, Piscataway, USA).

### Protein sequence and domain analysis

The protein sequence of WTK7-TM was predicted on the basis of the conserved domain database from the National Center for Biotechnology Information (NCBI) (https://www.ncbi.nlm.nih.gov/Structure/cdd/wrpsb.cgi) and SMART. Prediction of transmembrane helices was performed with TMHMM server v.2.0 (http://www.cbs.dtu.dk/services/TMHMM/), Phobious (https://phobius.sbc.su.se/) and Protter (http://wlab.ethz.ch/protter/#). The three-dimensional structure of WTK7-TM was predicted using AlphaFold v2.0[39]. The structural graphic was generated using PyMOL (The PyMOL Molecular Graphics System, v.2.0; Schrödinger).

### Phylogenetic analysis

Phylogenetic analysis was conducted to compare putative kinase domains including 184 putative kinase or pseudokinase domains used in the study ofWTK1[24], and 12 putative kinase or pseudokinase domains of WTK2[25], WTK3[15], WTK4[28], WTK5 (*Sr62*)[26], WTK6-vWA (*Lr9*)[27], and WTK7-TM. ClustalW in the MEGA was used for multiple alignment using the default setting and phylogenetic construction. Phylogenetic trees were built with the Neighbor-joining method in the MEGA and polished by Itol (https://itol.embl.de/).

### Functional marker development

Based on the genomic sequence of the *WTK7-TM* gene, we developed a specific primer, *WTK7-TM-FM*, to identify the presence of the *WTK7-TM* functional allele among *Triticeae* species. The dominant functional marker *WTK7-TM-FM* amplified a 1198 bp PCR product if the plant carrying *WTK7-TM* gene, otherwise no amplification product. A 2 × Taq Master Mix (Vazyme, China) was used in a total volume of 10 μL with the following amplification program: 95 °C for 3 min, 40 cycles of 95 °C for 15 s, 60 °C for 15 s and 72 °C for 30 s, 72 °C for 10 min. PCR products of functional marker *WTK7-TM-FM* were mixed with 2 μL of loading buffer (98% formamide, 10 mM EDTA, 0.25% bromophenol blue, and 0.25% xylene cyanol) and separated on 1% agarose gels. The coding sequence (CDS) of the *WTK7-TM* gene was amplified from 38 WEW accessions, which were tested carrying *WTK7-TM* gene by functional marker *WTK7-TM-FM*, using the primers listed in Supplementary Data 2. ClustalW was used for multiple sequence alignment.

### Primers

Primers used in this study are listed in Supplementary Data 2.

### Reporting summary

Further information on research design is available in the Nature Portfolio Reporting Summary linked to this article.

## Data availability

Data supporting the findings of this work are available within the paper and its Supplementary Information files. Transcriptome data generated from the current study are publicly available from the Genome Sequence Archive in the National Genomics Data Center, China National Center for Bioinformation/Beijing Institute of Genomics, Chinese Academy of Sciences at the accession CRA009664. The sequence of *Pm36* was deposited in the National Center for Biotechnology Information (NCBI) under the accession number OQ361691. Source data are provided in this paper.

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

## Acknowledgements

We are grateful to Profs. Qixin Sun and Tsomin Yang, China Agricultural University, for their advices and support during the research. Many thanks to Drs. Xueyong Zhang, Xianchun Xia, and Lihui Li of the Chinese Academy of Agricultural Sciences, Beijing, China for providing the Chinese wheat mini-core collection (MCC), worldwide wheat landrace collection, and Chinese wheat landraces, respectively; Dr. Antonio Blanco of the University of Bari Aldo Moro, Bari, Italy for providing resistant line 5BIL-29. This research was financially supported by the National Key Research and Development Program of China (2021YFA1300703 to Z.Y.L.), National Natural Science Foundation of China (32101735 to M.M.L. and U21A20224 to Z.Y.L.), Key Research and Development Program of Hebei Province (22326305D to C.G.Y) and Hainan Seed Industry Laboratory (B21HJ0111 to Z.Y.L.).

## Author contributions

Z.L., M.L. and Y.Z. designed the study. M.L., H.Z., H.X., K.Z., W.S., D.Z., Y.W., L.Y., Q.W., Y.C., D.Q., P.L., B.L., L.D., W.L., X.C., L.L., X.T. and C.Y. performed the experiments, conducted fieldwork, analyzed data, performed *Bgt* inoculation, and developed EMS mutants. L.L.D., J.X. and G.G. performed bioinformatic analysis. Y.L., D.Y., E.N., T.F. and H.L. provided powdery mildew resistance phenotyping, germplasm, or scientific support and advice. Z.L., M.L. and Y.Z. wrote the manuscript. All authors contributed to the article and approved the submitted version.

## Competing interests

The authors declare no competing interests.
