## [Peer Review File · Nature Communications]

A membrane associated tandem kinase from wild emmer wheat confers broad-spectrum resistance to powdery mildewReviewers' Comments:

Reviewer #1:

Remarks to the Author:

This paper describes the map-based cloning of the Pm36 gene. There has been a lot of work done and this new gene undoubtedly represents an interesting target to improve wheat resistance to powdery mildew and to possibly identify a new resistance mechanism. However, I have some criticisms. At many places in the document, it is referred that Pm36 is present in genotypes 3D232 and 5BIL-29 but you can't say that until you have the gene sequences from both accessions. Therefore, all along the document, there is a mix-up of the different locus from 3D232 and 5BIL-29, which makes it confusing. The cloning approach was very risky to validate CYB561-Domon and TM9SF4 genes without having a detailed view of the gene content of the resistant accession. It is pity that the same experiment (stable transformation) was not performed for WTK7 as for me it does not allow to fully conclude on the identification of Pm36. Indeed, the VIGS and tilling data demonstrate that WTK7 is involved in resistance but it is not the ultimate proof to say this is the only gene involved in resistance especially as 6 mutants are susceptible but do not contain mutations in the WTK7 gene. Furthermore, if the originality of the paper relies on the presence of a transmembrane domain in this new TKP, I would recommend the authors to look at the subcellular localization of the protein using GFP fusion for instance. The absence of WTK7 in cultivated wheat most likely explains its broad-spectrum resistance. I would be curious to know the resistance status and deployment of the other Pm genes and to have a paragraph in the discussion on how to use this gene in cultivated wheat without the risk to have it overcome.

There are several missing data reported below:

Introduction

L59 they are more recent and trough studies on wheat yield losses throughout the world. Savary et al. 2019. Please use this reference.

Please provide more detail about the pm genes?

L63 references 4 is not adapted

L103 The only way to say that MI3D232 is allelic to Pm36 is to have both genes cloned, which was not the case at the beginning of your study. The previous studies strongly suggest that they are allelic?

L108 please provide the type of markers

L113, please provide the exact physical size of the intervals for each genomes

L123: In Supplementary Fig.4 you provide an agarose gel (I guess) to evaluate the expression of your genes whereas in your Mat and Meth section it is written that you used the 2delta delta Ct method. Could you please result from this analysis instead of an agarose gel?

L124, How both gene were compared between 3D232 and Xueza0, this is not explained anywhere, which kind of polymorphisms was observed?

L129, Have you check the expression of your candidate genes in Fielder?

L133 As you are working with 5BIL-29, you cannot say that you are isolating the MI3D232 gene as you are not sure that these two accessions carry the same gene. Please correct this in the entire paragraph.

L146 9 or 8 additional genes compared ot CS?

L160 In Supplementary Table 4 you provide gene expression data of all the genes identified from 5BIL-29 in genotype MI3D232 but you have no idea if the same gene are present in MI3D232, please correct or prove that the MI3D232 is exactly the same as the one from 5BIL-29. Same remark for Supplementary Fig9

L166 I see 12 exons from your figure?

L172 How can you say that the WTK7-TM is up regulated after infection with isolate E09 as you do not present data of plant without infection?

L185 I wouldn't say here that WTK7 is Pm36 as it could be at this stage of your experiments two genes.

L190, Which gene sequence, the CDS?

L198 not the right figure cited.

L204 You can't say that it is Pm36 gene. It is most likely Pm36 but this need to be validated by its transfer in a susceptible background.

L 222 Where are the result of the auto-phosphorylation study?

L229 How the presence of WTK7 was checked?

L412 no "s" ate "table"

L435 Supplementary Table 3?

L441 Supplementary Table 3?

L460, How many leaves were used for RT-PCR and RNA-seq, please provide more information about the RT and PCR conditions.

L463 Please provide further detail about samples used for RNA-seq analysis, 24hpi but inoculated with which isolate and mock as well?

L474, it is sometimes confusing so please provide further detail about your biological and technical replicates.

L492 Please detail how the fragment were cloned

L497 at 5 or 12 ? because the virus load must be quite different between both timing.

L498 Please provide the reference of the celite

L516 Please provide the set of SSR markers used.

L518 Which method was used to get the sequence? I can't find the primers used in Supplementary Table 1. It is important to know the exact sequence obtained to understand why 6 mutants are susceptible without mutation in WTK7.

L524 Supplementary Table 1?

L531 Supplementary Table 1?

Supplementary Fig 1 could you please provide information about the isolates? Please provide the scores for each interaction.

Supplementary Fig.2 is not described in the main text and there is no Mat and Meth section relative to this experiment. How was the adult stage resistance assessed?

Supplementary Fig.3 What are RBH, SBH? What is the scale? Is WEWSeq v1.0 Zavitan?

Supplementary Fig.6 I guess the number above the TE represents their insertion time but it does not range from 0.22 to 2.5 MYA as indicated in the legend? I'm not sure how the timing of the two duplication was estimated as the the insertion time of TE for both duplications overlap largely? Please provide further details.

Supplementary Fig.7 Please provide explanation on how the figure were obtained.

Supplementary Fig.8 According to the different software used the kinase domain are either predicted to be outside of the cell or inside. Could you please clarify this uncertainty?

Supplementary Fig.8 Please provide in the legend how the proportion of each transcript was estimated and from which samples.

Supplementary Fig.11 What is the marker lane? Please mention if any statistical test was apply to evaluate significant differences in expression.

Supplementary Fig.14 Could you please indicate how the alignment was done, was are the different colors? Could you please indicate as well the 8 conserved residues of active kinase as mentioned in Klymiuk et al. 2021.

Supplementary Fig.15 Please provide the exact protocol for this WTK7 diagnostic marker.

Supplementary Table 1, Please indicate which markers are SSR and dCAP and provide which enzyme

was used for the dCAP markers.

Supplementary Table 2: Why the physical size of the interval is between WGGBM4 and WGGBM9 while in the text it is between WGGBM4 and 7. Could you please indicate the size of the interval for the other wheat sequenced accessions as indicated in the text (Ref 26 and 27).

Could you please provide names of genetic resource centres where wheat resources are available?

Mat and Meth section: Please provide how many plants were evaluated and the number of replicates. PCR conditions for genotyping?

Reviewer #2:

Remarks to the Author:

Li et al. reported the cloning of Pm36, a unique membrane associated tandem kinase (WTK7-TM), by combination of PacBio SMRT long-read genome sequencing and map-based cloning. The gene was validated by mutants and VIGS, and the proof is solid. Pm36 could be only found in the wild emmer wheat populations and showed resistance to 104 Bgt isolate. Pm36 has been introgressed into durum and common wheat background showing great gain in wheat resistance breeding. The authors did some mechanism research, but the story is not clear and insufficient. Some results are different with previous publication which should be carefully discussed or repeat to be suitable for publication. Please see 'Nirmala, J., Drader, T., Chen, X., Steffenson, B. and Kleinhofs, A., 2010. Stem rust spores elicit rapid RPG1 phosphorylation. *Molecular plant-microbe interactions*, 23(12), pp.1635-1642'. In this paper, RPG1 protein is phosphorylated within 5 min by exposure to spores from avirulent but not virulent races of stem rust. The pseudokinase pK1 domain is required for disease resistance but not phosphorylation.

Questions/suggestions

1. Line 80: ref19: has been published on nature plants now, update the citation.
2. What is the subcellular localization of WTK-TM, since it contains a transmembrane domain, did you have some proof as the TM domain located on the membrane as shown in Fig.1e?
3. It is better to check ROS and cell death response of WTK7-TM. Is the intraROS play role for the WTK7-TM resistance? Follow this paper, <https://apsjournals.apsnet.org/doi/10.1094/PHYTO-07-22-0271-FI>
4. Line 210 and 258: What is the unique resistance mechanism or different resistance mechanism for the WTK7-TM? If you do not have a clear proof, please tone down this part.
5. Have you checked the entire protein WTK7-TM has the autophosphorylation or not? I surmised the kinase domain alone may has autophosphorylation, the entire WTK7-TM may have kinase activity after reorganization with effector.
6. Have you check the different WTK7-TM alleles among the published wheat reference genomes? Are there some homologues, orthologs, paralogs? Could the insertion part sequence be original from other location of wheat chromosome, or original from other wheat relatives, or it is just totally a new sequence?
7. Did you have some information (collect locations or wheat lines) on the 104 Bgt isolates, references or Table? Are they present as enough genetic diversity?

8. All the figure legends of supplementary Figures should be improved. Such as, add lactation information in Sup Fig.2. How did you do Bgt infection in field or just nature infection? Add the gene information on supp Fig.3. and the main legend should be short no need to add the website address. The supp Fig.4 should be "Genes expression analysis in the M13D232 genetic region." Supp. Fig. 5 "transgenic complementation" may be used for overexpression functional genes in the recessive mutations. Supp Fig.6 - How do you know the original segment, duplication 1 and 2, and the MYA years of each duplication and each gene? Supp. Fig.7. Which version of the reference are used? Add how do you do the analysis. Supp Fig.9 add information what reference you used, the long-read assembly contigs? Supp. Fig. 10. How do you calculate the different transcriptional isoforms? Supp. Fig 12 a, two should be Two. The title legend of Fig 13 is not good, and the protein structure is not clear, why you put the Rgp1 and Pm36 together? You need to add the PCR product size information in the Fig. 15, and you did not show how you develop the specific markers? What is the meaning of Pm36 in the Fig. 15? You need to write where you get this map in Fig. 19? Google map or ? Please consider to add the country name of Israel in the map since the most WEW is original from here.
9. Supp. Fig.11. I think it is better to add mock infection (non-infection) as control. WTK1 expression is reduced after Pst infection. While your result is different and need to be careful.
10. You need to give the Kinase I and II a family name as previous WTK genes, such as LRR-8B or WAK?
11. The supp. Table should be Excel format.

Reviewer #3:

Remarks to the Author:

In their study the authors describe the identification and isolation of a novel powdery mildew resistance gene originating from wild emmer wheat. For that, they sequenced the carrier, a tetraploid WEW-durum introgression line. I will focus my review on the genomics work carried out in this study, as the main area of my expertise.

I have a few points that require clarification and/or some more work:

- A.) The genome assembly statistics, contiguity in particular, look good but not great. I understand the genome assembly was left in scaffold state, which is reasonable for the main objectives of this study (identification of the resistance gene). Nevertheless, I see a missed opportunity here, too. Advancing the genome assembly to a pseudomolecule level would not only make this a much more valuable resource for downstream and follow-up analyses, but it would also set the genomic region harboring the novel resistance gene into a full genomic context. This could be of relevance for assessing gene regulation in a (epi-) genomic context, as well as chromosome organization. I would like to suggest the authors to consider doing Hi-C and bringing the assembly to a full pseudo-chromosome level, which could be performed with very reasonable efforts and costs.
- B.) While synteny/collinearity analyses have been performed with other genomes/species, I'm missing a proper comparison of the resistance carrying region with the homologous region on the other WEW subgenome. Supplementary figure 3 needs a complete makeover in my view, I do not intuitively understand what is displayed here. It is also missing any explanatory figure legend.
- C.) Gene prediction and annotation: the methods detail that FgenesH and BlastP was used to identify genes in the respective genomic region. To understand what was done here, and to assess the prediction quality, this part needs more explanations and details. What database was blasted against? How were the final gene structures modeled? What was the impact of the transcriptome data on gene predictions? These are very critical points as many of the downstream analysis dealing with the novel resistance gene actually depend on this gene structure prediction.
- D.) The authors describe transcriptional variants of the resistance gene (Supp Figure 10), but I'm not clear where these actually come from and what the supporting evidence for that distribution is...did the

authors also utilize long-read expression data (Iso-Seq) to validate those structures?

E.) Supp Figure 9: it is hard to see differences in this screenshot, and a figure legend is missing as well. I suggest to re-do this figure.

Point-by-point response to Reviewers

Reviewer #1 (Remarks to the Author):

This paper describes the map-based cloning of the *Pm36* gene. There has been a lot of work done and this new gene undoubtedly represents an interesting target to improve wheat resistance to powdery mildew and to possibly identify a new resistance mechanism.

Response: We are very grateful to your comments for the manuscript.

However, I have some criticisms. At many places in the document, it is referred that *Pm36* is present in genotypes 3D232 and 5BIL-29 but you can't say that until you have the gene sequences from both accessions. Therefore, all along the document, there is a mix-up of the different locus from 3D232 and 5BIL-29, which makes it confusing.

Response: Thanks for your suggestions. Follow your advices, we modified the description of *Pm36* and *MI3D232* gene in the revised manuscript and added research background to clarify the relationship of *Pm36* and *MI3D232* genes in the introduction part. "*Pm36*, a wild emmer wheat (WEW)-derived powdery mildew resistance gene was identified and mapped on chromosome arm 5BL in the tetraploid WEW-durum introgression line 5BIL-29 (Blanco et al, 2008). We previously mapped a powdery mildew resistance gene *MI3D232* derived from WEW in the hexaploid wheat introgression line 3D232 (Zhang et al, 2010) on the overlapping genetic interval with *Pm36* and deduced that they are likely the same gene or alleles by comparative genetic linkage maps (Zhang et al, 2015)."

Reference

1. Blanco, A. et al. Molecular mapping of the novel powdery mildew resistance gene *Pm36* introgressed from *Triticum turgidum* var. *dicoccoides* in durum wheat. *Theor. Appl. Genet.* **117**, 135–142 (2008).
2. Zhang, H. et al. Genetic and comparative genomics mapping reveals that a powdery mildew resistance gene *MI3D232* originating from wild emmer co-segregates with an NBS-LRR analog in common wheat (*Triticum aestivum* L.). *Theor. Appl. Genet.* **121**, 1613–1621 (2010).
3. Zhang, D. et al. Comparative genetic mapping revealed powdery mildew resistance gene *MIWE4* derived from wild emmer is located in same genomic region of *Pm36* and *MI3D232* on chromosome 5BL. *J. Integr. Agric.* **14**, 603–609 (2015).

The cloning approach was very risky to validate *CYB561-Domon* and *TM9SF4* genes without having a detailed view of the gene content of the resistant accession. It is pity that the same experiment (stable transformation) was not performed for WTK7 as for me it does not allow to fully conclude on the identification of *Pm36*. Indeed, the VIGS and tilling data demonstrate that WTK7 is involved in resistance but it is not the ultimate proof to say this is the only gene involved in resistance especially as 6 mutants are susceptible but do not contain mutations in the WTK7 gene.

Response: Thanks for the comments. In the revised version, we added the sequence information of *CYB561-Domon* (Supplementary Fig. 4) and *TM9SF4* (Supplementary Fig.

5) genes. To answer the reviewer's concerns, we performed stable transformation experiments for *WTK7-TM* and demonstrated that the T₁ positive transgenic lines of *WTK7-TM* showed immune to *Bgt* isolate E09 (Fig. 3). Based on the results of VIGS, mutagenesis and stable transgenic assays, we confirmed that *WTK7-TM* is *MI3D232/Pm36*.

Furthermore, if the originality of the paper relies on the presence of a transmembrane domain in this new TKP, I would recommend the authors to look at the subcellular localization of the protein using GFP fusion for instance.

Response: Thanks for your suggestions. To investigate the subcellular localization of *WTK7-TM*, we built recombinant plasmid *ProUbi:WTK7-TM-GFP* to express *WTK7-TM* protein in wheat leaf mesophyll protoplasts. We found that *WTK7-TM* protein is localized in the nucleus, cytomembrane and cytoplasm in the wheat leaf mesophyll protoplasts (Supplementary Fig. 16). We have updated this information in the revised manuscript (lines 266-268).

The absence of *WTK7* in cultivated wheat most likely explains its broad-spectrum resistance. I would be curious to know the resistance status and deployment of the other *Pm* genes and to have a paragraph in the discussion on how to use this gene in cultivated wheat without the risk to have it overcome.

Response: Thank you for your advices. To date, more than a hundred *Pm* genes have been reported, a few of them can provide resistance to most or all *Bgt* isolates, such as *Pm12* from *Aegilops speltoides* (Zhu et al, 2023), *Pm21* from *Dasypyrum villosum* L. Candagy (He et al, 2018; Xing et al, 2018), and *Pm24* from common wheat (Lu et al, 2020). *Pm38/Yr18/Lr34/Sr57* (Krattinger et al, 2009) and *Pm46/Yr46/Lr67/Sr55* (Moore et al. 2015) provide durable resistance to powdery mildew, leaf rust, stem rust, and stripe rust. *Pm21* confers immunity to all tested *Bgt* isolates and has been widely utilized in wheat breeding in China since 1995 (Chen et al, 2013). Yet, *Bgt*-resistant wheat cultivars carrying single *Pm* gene are prone to lose their resistance due to strong pathogen selective pressure. Therefore, combining multiple and diverse resistance genes is a sustainable approach to disease management in wheat breeding. We have updated this information in the Discussion of the revised manuscript (lines 422-429 in Discussion).

Reference

1. Zhu, S. et al. Orthologous genes *Pm12* and *Pm21* from two wild relatives of wheat show evolutionary conservation but divergent powdery mildew resistance. *Plant Commun.* **4**, 100472 (2023).
2. He, H. et al. *Pm21*, encoding a typical CC-NBS-LRR protein, confers broad-spectrum resistance to wheat powdery mildew disease. *Mol. Plant* **11**, 879–882 (2018).
3. Xing, L. et al. *Pm21* from *Haynaldia villosa* encodes a CC-NBS-LRR protein conferring powdery mildew resistance in wheat. *Mol. Plant* **11**, 874–878 (2018).
4. Lu P, et al. A rare gain of function mutation in a wheat tandem kinase confers resistance to powdery mildew. *Nat. Commun.* **11**, 680 (2020).
5. Krattinger, S. G. et al. A putative ABC transporter confers durable resistance to multiple fungal pathogens in wheat. *Science* **323**, 1360–1363 (2009).

6. Moore, J. W. et al. A recently evolved hexose transporter variant confers resistance to multiple pathogens in wheat. *Nat. Genet.* **47**, 1494–1498 (2015).

There are several missing data reported below:

Introduction

L59, they are more recent and trough studies on wheat yield losses throughout the world. Savary et al. 2019. Please use this reference.

Response: Thank you for your advice. We have modified the reference in the revised manuscript (line 58).

Reference

1. Savary, S. et al. The global burden of pathogens and pests on major food crops. *Nat. Ecol. Evol.* **3**, 430–439 (2019).

Please provide more detail about the pm genes?

Response: Thank you for your advice. In the revised version, we have provided research progress of *Pm* genes in the Introduction (lines 66-79).

L63, references 4 is not adapted

Response: Thanks. In the revised manuscript, we have changed the reference 4.

L103, The only way to say that *MI3D232* is allelic to *Pm36* is to have both genes cloned, which was not the case at the beginning of your study. The previous studies strongly suggest that they are allelic?

Response: Thanks for your suggestions. “*Pm36*, a wild emmer wheat (WEW)-derived powdery mildew resistance gene was identified and mapped on chromosome arm 5BL in the tetraploid WEW-durum introgression line 5BIL-29³⁰. We previously mapped a powdery mildew resistance gene *MI3D232* derived from WEW in the hexaploid wheat introgression line 3D232³¹ on the overlapping genetic interval with *Pm36* and deduced that they are likely the same gene or alleles by comparative genetic linkage maps³².”

Reference

1. Blanco, A. et al. Molecular mapping of the novel powdery mildew resistance gene *Pm36* introgressed from *Triticum turgidum* var. *dicoccoides* in durum wheat. *Theor. Appl. Genet.* **117**, 135–142 (2008).

2. Zhang, H. et al. Genetic and comparative genomics mapping reveals that a powdery mildew resistance gene *MI3D232* originating from wild emmer co-segregates with an NBS-LRR analog in common wheat (*Triticum aestivum* L.). *Theor. Appl. Genet.* **121**, 1613–1621 (2010).

3. Zhang, D. et al. Comparative genetic mapping revealed powdery mildew resistance gene *MIWE4* derived from wild emmer is located in same genomic region of *Pm36* and *MI3D232* on chromosome 5BL. *J. Integr. Agric.* **14**, 603–609 (2015).

L108, please provide the type of markers

Response: Thanks. We have updated this information in the revised manuscript (lines 126-127, 131-133)

L113, please provide the exact physical size of the intervals for each genome

Response: Thanks for your suggestions. We modified this part in line 134 in the revised version. The exact physical size of the intervals for each genome have provided in new Supplementary Fig. 7.

L123, In Supplementary Fig.4 you provide an agarose gel (I guess) to evaluate the expression of your genes whereas in your Mat and Meth section it is written that you used the $2\Delta\Delta$ Ct method. Could you please result from this analysis instead of an agarose gel?

Response: Thank you for your advice. As shown in new Supplementary Fig. 6, we have examined the expression level of *CYB561-Domon* and *TM9SF4* by quantitative reverse transcription PCR (qRT-PCR).

L124, How both gene were compared between 3D232 and Xueza0, this is not explained anywhere, which kind of polymorphisms was observed?

Response: Thanks for your suggestions. We cloned *CYB561-Domon* and *TM9SF4* genes from 3D232 and XZ. Sequence alignment of *CYB561-Domon* gene from lines 3D232 and Xueza0 (XZ) showed in Supplementary Fig. 4. Sequence alignment of *TM9SF4* gene from lines 3D232 and Xueza0 (XZ) showed in Supplementary Fig. 5. SNPS and InDels were observed for *CYB561-Domon* and SNPS were found for *TM9SF4* between 3D232 and XZ.

L129, Have you check the expression of your candidate genes in Fielder?

Response: Thank you for your advice. We have checked and found the expression of *CYB561-Domon* and *TM9SF4* genes in the Fielder and T₁ positive transgenic plants of *ProUbi:CYB561-Domon* and *ProUbi:TM9SF4*, respectively. We added the information in the Supplementary Fig. 6 in the revised manuscript.

L133, As you are working with 5BIL-29, you cannot say that you are isolating the *MI3D232* gene as you are not sure that these two accessions carry the same gene. Please correct this in the entire paragraph.

Response: Thanks for your suggestions. We modified this part in Lines 168–193 in the revised version.

L146, 9 or 8 additional genes compared to CS?

Response: Thank you for your careful review. 8 and 7 additional genes compared to CS and Zavitan, respectively. It has been corrected in the revised manuscript (Line179-182).

L160, In Supplementary Table 4 you provide gene expression data of all the genes identified from 5BIL-29 in genotype *MI3D232* but you have no idea if the same gene are present in 3D232, please correct or prove that the *MI3D232* is exactly the same as the one from 5BIL-29. Same remark for Supplementary Fig9

Response: Thank you for your advices. It has been corrected in Supplementary Table 5 in the revised manuscript.

L166, I see 12 exons from your figure?

Response: Thank you for your careful review. It has been corrected in the revised manuscript (Line 204)

L172, How can you say that the *WTK7-TM* is up regulated after infection with isolate E09 as you do not present data of plant without infection?

Response: We added a mock treatment in *WTK7-TM* expression assay. We found that *WTK7-TM* is constitutively expressed in the root, stem, and leaf tissues without *Bgt* infection (Supplementary Fig. 12a), but could be slightly affected after *Bgt* isolate E09 infection in lines 3D232 and 5BIL-29 (Supplementary Fig. 12b). We corrected it as suggested in lines 206-211.

L185, I wouldn't say here that WTK7 is *Pm36* as it could be at this stage of your experiments two genes.

Response: Thank you for your careful review. We corrected it as suggested.

L190, Which gene sequence, the CDS?

Response: We checked the full-length genomic sequence of the *WTK7-TM* including the promoter region, gene coding region, and terminator region from each of the 23 mutants and found 17 mutants carried mutations within the *WTK7-TM* gene CDS sequence, including four premature stop codons, one frameshift alternative splicing variant, and 12 missense mutations (Fig. 2d and Supplementary Table 6).

L198, not the right figure cited.

Response: Thank you for your careful review. It has been corrected.

L204, You can't say that it is *Pm36* gene. It is most likely *Pm36* but this need to be validated by its transfer in a susceptible background.

Response: Thanks for your suggestions. We have corrected the sentence in Line 247-249. The function of *WTK7-TM* for resistance to *Bgt* was confirmed by transgenic assay in lines 250-259 in Result part.

L222, Where are the result of the auto-phosphorylation study?

Response: Thank you for your careful review. In Fig 5c, the up panel is autoradiograph image indicating the phosphorylation state and the bottom panel is the SDS-PAGE gel stained with coomassie brilliant blue indicating the proteins used in the reaction. In *in vitro* phosphorylation experiment, we respectively incubated recombinant proteins MBP-WTK7-TM^{KinI} (about 71 kDa), MBP-WTK7-TM^{KinII} (about 75 kDa), MBP-WTK7-TM^{KinI-KinII} (about 105 kDa) and MBP-WTK7-TM-FL (about 116.7 kDa) without substrate in kinase buffer with [γ -³²P]-ATP and found that only MBP-WTK7-TM^{KinI-KinII} has a weak autoradiograph signal indicated by arrow located in 105 kDa in the up panel (Fig. 5b, c and Supplementary Fig.

17). And when we added the general phosphorylation substrate myeline basic protein in the reaction, only the MBP-WTK7-TM^{KinI-KinII} can phosphorylate the myeline basic protein showing the strongest signal band located in 17 kDa and meanwhile show a little stronger autoradiograph signal (Fig. 5b, c and Supplementary Fig. 17). By contrast, when MBP-WTK7-TM^{KinI-KinII} being incubated with GST-GFP (as a negative control), it can't phosphorylate GST-GFP but still has the autoradiograph band in the last lane.

L229, How the presence of *WTK7* was checked?

Response: “Based on the genomic sequence of *WTK7-TM* gene, we developed a specific primer, *WTK7-TM-FM*, to identify the presence of the *WTK7-TM* functional allele among *Triticeae* species. The dominant functional marker *WTK7-TM-FM* amplified a 1,198 bp PCR product if the plant carrying *WTK7-TM* gene, otherwise no amplification product. A 2 × Taq Master Mix (Vazyme, China) was used in a total volume of 10 μL with the following amplification program: 95 °C for 3min, 40 cycles of 95 °C for 15 s, 60 °C for 15 s and 72 °C for 30 s, 72 °C for 10min. PCR products of functional marker *WTK7-TM-FM* were mixed with 2 μl of loading buffer (98% formamide, 10mM EDTA, 0.25% bromophenol blue, and 0.25% xylene cyanol) and separated on 1% agarose gels.” We added the functional marker development information in Lines 687-696 in M&M.

L412, no “s” ate “table”

Response: Thanks for your careful review. It has been corrected.

L435, Supplementary Table 3?

Response: Thanks for your careful review. It has been corrected.

L441, Supplementary Table 3?

Response: Thanks for your careful review. It has been corrected.

L460, How many leaves were used for RT-PCR and RNA-seq, please provide more information about the RT and PCR conditions.

Response: Thanks for your advices. For qRT-PCR experiment, three biological replicates and three technical replicates for each sample at 0, 6, 12, 24, 36, 48, and 72 hpi with *Bgt* isolate E09 were performed. We used three leaves from three plants representing three biological replicates at each time point and conducted each biological replicate three times representing three biological replicates. For RNA-seq experiment, the RNA samples from two-leaf stage seedling leaves of 3D232 or 5BIL-29 inoculated with *Bgt* isolate E09 at 24 hpi were selected to construct cDNA libraries and subjected to high-throughput paired-end sequencing by Illumina HiSeq2000. The information was updated in the revised manuscript.

L463, Please provide further detail about samples used for RNA-seq analysis, 24 hpi but inoculated with which isolate and mock as well?

Response: Thanks for your advices. In order to qualitative analysis of candidate genes expression in *Pm36* locus, we conducted the RNA-Seq experiment. Three leaves from three plants of 3D232 at two-leaf stage seedling inoculated with *Bgt* isolate E09 at 24 hpi

were collected and mixed for constructing cDNA libraries and subjected to high-throughput paired-end sequencing by Illumina HiSeq2000. Mock treatment was not conducted in RNA-seq experiment. The information was updated in the revised manuscript.

L474, it is sometimes confusing so please provide further detail about your biological and technical replicates.

Response: Thanks for your advices. For qRT-PCR experiment, three biological replicates and three technical replicates for each sample at 0, 6, 12, 24, 36, 48, and 72 hpi with *Bgt* isolate E09 were performed. We used three leaves from three plants representing three biological replicates at each time point and conducted each biological replicate three times representing three biological replicates.

L492, Please detail how the fragment were cloned

Response: Thanks for your suggestions. Two fragments of *WTK7-TM*, one from the Kin I domain and the other from the Kin II domain, were amplified from recombinant vector pEASY:*WTK7-TM TV1* carrying *TV1* transcription isoform of *WTK7-TM* using primers *WTK7-TM-VIGS-1* and *WTK7-TM-VIGS-2* respectively and inserted into pCa- γ BLIC to generate independent recombinant constructs, BSMV:*WTK7-TM^{Kin I}*, and BSMV:*WTK7-TM^{Kin II}*, respectively. The information was updated in the revised manuscript.

L497 at 5 or 12? because the virus load must be quite different between both timing.

Response: Thanks for your comments. At 10 dpi, the infiltrated *N. benthamiana* leaves were ground in 20 mM Na-phosphate buffer (PH7.2) containing 1% celite (Yuan et al, 2011), and the sap was mechanically inoculated onto 3D232 and 5BIL-29 leaves at the two-leaf stage. The information was updated in the revised manuscript Lines 590-592.

Reference

Yuan, C. et al. A high throughput barley stripe mosaic virus vector for virus induced gene silencing in monocots and dicots. *PLoS One* **6**, e26468 (2011).

L498, Please provide the reference of the celite

Response: Thanks for your suggestions. The reference of the celite has been added in the revised manuscript (line 591).

Reference

Yuan, C. et al. A high throughput barley stripe mosaic virus vector for virus induced gene silencing in monocots and dicots. *PLoS One* **6**, e26468 (2011).

L516, Please provide the set of SSR markers used.

Response: Thanks for your advices. For authenticity of mutant genotype analysis, a set of SSR markers were used and the sequence information of SSR markers were added in Supplementary Table 2.

L518, Which method was used to get the sequence? I can't find the primers used in

Supplementary Table 1. It is important to know the exact sequence obtained to understand why 6 mutants are susceptible without mutation in WTK7.

Response: Thanks for your advices. The full-length genomic sequence of the *WTK7-TM* including the promoter region, gene coding region, and terminator region from each of the 23 mutants were obtained using the primers, WTK7-60, WTK7-61 and WTK7-62 listed in Supplementary Table 2 and compared with the wild-type 3D232 using DNAMAN 8 software.

L524, Supplementary Table 1?

Response: Thanks. It has been corrected.

L531, Supplementary Table 1?

Response: Thanks. It has been corrected.

Supplementary Fig 1 could you please provide information about the isolates? Please provide the scores for each interaction.

Response: Thanks for the suggestions. We have added the information about the isolates and the scores for each interaction in Supplementary Table 1.

Supplementary Fig.2 is not described in the main text and there is no Mat and Meth section relative to this experiment. How was the adult stage resistance assessed?

Response: Thank you for pointing out this issue. In the revised manuscript, we have added the detail of the adult stage resistance assessment in M & M (Lines 469-472). Powdery mildew reactions evaluated at the grain-filling stage were recorded as the seedling stage resistance assessment to *Bgt* isolates in M & M (Lines 469-472).

Supplementary Fig.3 What are RBH, SBH? What is the scale? Is WEWSeq v1.0 Zavitan?

Response: Thanks. RBH and SBH are abbreviation of reciprocal best hit and single-side best hit, respectively (Chen et al, 2020). This diagram is generated based on the actual each genome sequence size using Triticeae GeneTribe (wheat.cau.edu.cn/TGT/). We used WEWSeq v1.0 Zavitan reference genome sequence for collinearity analysis of *Pm36* locus. To better demonstrate the results of comparative genomic analyses, we modified the Supplementary Fig. 3 based on the old version and you can find the new Supplementary Fig. 7.

Reference

Chen, Y. et al. A collinearity-incorporating homology inference strategy for connecting emerging assemblies in the Triticeae Tribe as a pilot practice in the plant pangenomic Era. *Mol. Plant.* 13, 1694-1708 (2020).

Supplementary Fig.6 I guess the number above the TE represents their insertion time but it does not range from 0.22 to 2.5 MYA as indicated in the legend? I'm not sure how the timing of the two duplication was estimated as the insertion time of TE for both duplications overlap largely? Please provide further details.

Response: Many thanks for pointing out our mistake in the insertion time of the new

Supplementary Fig. 9 and its Figure legends, we corrected it in this manuscript. For the estimation of TEs' insertion time, first, we used Repeat Sequence Database (TREP) (<https://trep-db.uzh.ch/>) to identify known repetitive DNA. And then, the long terminal repeat sequences of LTR retrotransposable elements were delineated using Dotter analysis and manually checked. The evolutionary distance of the two LTR sequences was estimated by MEGA11 (<https://www.megasoftware.net/>) using a substitution rate of 1.3×10^{-8} mutations per site per year. For the duplication of the fragments, fragment 1 and fragment 2 are 92 kb and 69 kb, respectively. Except the large insertion/deletion (about 23 kb), fragment 1 and fragment 2 are only different in SNPs and small insertion/deletion. We aligned the sequences of fragment 1 and fragment 2 and then calculate their divergence time by MEGA11 as above mentioned. As for the original genomic region, there were only gene sequences (F-box and serpin) and a partial of repetitive sequences were conserved. We use these sequences to calculate their divergence time. However, It's hard to say which one is original and we only can calculate their divergence time. The result was revised in the revision.

Supplementary Fig.7 Please provide explanation on how the figure were obtained.

Response: The genomic sequences were obtained using markers *WGGBM2* and *WGGBM7* flanking the *MI3D232/Pm36* locus based on Chinese Spring RefSeqv1.0 reference genome, Zavitan WEW_v1.0 reference genome and 5BIL-29 contig ptg0004221: 5509738-6683671. Sequence similarity comparison was performed using YASS genomic similarity search tool (<https://bioinfo.cristal.univ-lille.fr/yass/yass.php>). a: Zavitan (vertical) vs 5BIL-29 (horizontal); b: Zavitan (vertical) vs Chinese spring (horizontal). We have updated the information in new Supplementary Fig 8.

Supplementary Fig.8 According to the different software used the kinase domain are either predicted to be outside of the cell or inside. Could you please clarify this uncertainty?

Response: Thanks for the comments. Indeed, we observed that the predicted cellular location of the kinase domain may be differences in the different software. This may be due to the different database libraries and models used. The predicted cellular location often needs to be verified by more experimental evidence. However, when same software was applied, we found the WTK7-TM is located outside of the membrane and all the other characterized WTKs are located inside of the membrane (Supplementary Fig 11, 15).

Supplementary Fig.10 Please provide in the legend how the proportion of each transcript was estimated and from which samples.

Response: Thank you for your advices. We have added the proportion of each transcript in the legend of Supplementary Fig.10. Cloning and sequencing of the *WTK7-TM* cDNA clones identified nine transcript variants, designated *TV1* to *TV9* from 5BIL-29. One hundred single colonies were sequenced using the Sanger sequencing. The numbers and percentages of variants are shown on the right, e.g., 59/100 (59%).

Supplementary Fig.11 What is the marker lane?

Response: Thanks for your suggestions. The marker lane is DNA ladder and we have

been corrected it in new Supplementary Fig.19.

Please mention if any statistical test was applied to evaluate significant differences in expression.

Response: Thank you for pointing out this issue. The *t*-test was used to evaluate significant differences in gene expression. We have corrected it in Supplementary Figs 6d, 6e, and 12b.

Supplementary Fig.14 Could you please indicate how the alignment was done, what are the different colors? Could you please indicate as well the 8 conserved residues of active kinase as mentioned in Klymiuk et al. 2021.

Response: Thanks for your suggestions. The kinase domains of WTK7-TM were used to do BLASTP search of the NCBI protein database for searching similar proteins. Eleven reported plant protein kinases, *Arabidopsis thaliana* Cerk1, *Arabidopsis thaliana* BIK1, *Zea mays* Pti1 protein, *Hordeum vulgare* ROP binding kinases 1, *Arabidopsis thaliana* Protein kinase PK, *Solanum lycopersicum* Pto, *Thinopyrum longatum* Potein kinase PK, *Oryza sativa* RIPK, *Arabidopsis thaliana* BSK1, *Arabidopsis thaliana* BRI1, *Dasypyrum villosum* Stpk-V, and tandem kinase proteins, Rpg1, Yr15 (WTK1), Sr60 (WTK2), Pm24 (WTK3), WTK4, Sr62 (WTK5), Lr9 (WTK6-vWA) and Pm36 (WTK7-TM) were selected for multiple alignment using Clustal Omega. Different colors indicate different degrees of Amino acid conservation. Red triangles represent the eight key residues of active kinase as shown in new Supplementary Fig.17 and as mentioned in Klymiuk et al. 2021.

Supplementary Fig.15 Please provide the exact protocol for this *WTK7* diagnostic marker.

Response: Thanks for your advices. "Based on the genomic sequence of *WTK7-TM* gene, we developed a specific primer, *WTK7-TM-FM*, to identify the presence of the *WTK7-TM* functional allele among *Triticeae* species. The dominant functional marker *WTK7-TM-FM* amplified a 1,198 bp PCR product if the plant carrying *WTK7-TM* gene, otherwise no amplification product. A 2 × Taq Master Mix (Vazyme, China) was used in a total volume of 10 µL with the following amplification program: 95 °C for 3min, 40 cycles of 95 °C for 15 s, 60 °C for 15 s and 72°C for 30 s, 72°C for 10min. PCR products of functional marker *WTK7-TM-FM* were mixed with 2 µl of loading buffer (98% formamide, 10mM EDTA, 0.25% bromophenol blue, and 0.25% xylene cyanol) and separated on 1% agarose gels." We added the functional marker development information in Lines 687-696 in M&M.

Supplementary Table 1, Please indicate which markers are SSR and dCAP and provide which enzyme was used for the dCAP markers.

Response: Thanks for your suggestions. We added the maker type as suggested in Supplementary Table 2. The dCAPS marker *WGGBM2* linked with *MI3D232* gene was developed with restriction enzyme *XhoI*. We corrected it as suggested in Supplementary Table 2.

Supplementary Table 2: Why the physical size of the interval is between *WGGBM4* and *WGGBM9* while in the text it is between *WGGBM4* and 7. Could you please indicate the

size of the interval for the other wheat sequenced accessions as indicated in the text (Ref 26 and 27).

Response: Thank you for your advices. *MI3D232* gene was narrowed down to a genetic interval of 0.021-cM flanked by the closest derived cleaved amplified polymorphic sequences (dCAPS) marker *WGGBM2* and SSR marker *WGGBM7*. The physical interval between markers *WGGBM2* and *WGGBM7* were used for gene annotation and comparative analysis. The information was corrected and updated in Supplementary Table 4. We added the physical sizes of the *MI3D232* locus corresponding to multiple reference genomes, including *Aegilops speltoides*, *Triticum dicoccoides*, *Triticum turgidum*, *Triticum spelta*, 10+ wheat Genome etc. as updated in new Supplementary Fig. 7.

Could you please provide names of genetic resource centres where wheat resources are available?

Response: Thanks. In our experiment, 442 wild emmer wheat (*T. turgidum* ssp. *dicoccoides*), 636 and 394 common wheat accessions from China and the other 34 countries were collected and preserved. Most of these accessions were obtained from the Wheat Genetic and Genomic Resources Center, Kansas State University, The USDA-ARS National Small Grains Collection, and University of Haifa Israel. The 262 entries of Chinese wheat mini-core collection (MCC) were available in National Key Facility for Crop Gene Resources and Genetic Improvement, Chinese Academy of Agriculture Sciences. The 125 cultivated emmer wheat (*T. turgidum* subsp. *dicoccum*), 230 durum (*T. turgidum* subsp. *durum*) has the official Plant ID, such as PI ×××, Ctr ××× as shown in Supplementary Table 7. If researchers need these wheat germplasm resources, they can also contact us in private.

Mat and Meth section: Please provide how many plants were evaluated and the number of replicates. PCR conditions for genotyping?

Response: Thanks. We used three leaves from three plants representing three biological replicates at each time point and conducted each biological replicate three times representing three biological replicates. We developed a functional marker *WTK7-TM-FM* for genotyping among various *Triticeae* species. The dominant functional marker *WTK7-TM-FM* amplified a 1,198 bp PCR product if the plant carrying the *WTK7-TM* gene, otherwise no amplification product. "A 2 × Taq Master Mix (Vazyme, China) was used in a total volume of 10 µL with the following amplification program: 95 °C for 3min, 40 cycles of 95 °C for 15 s, 60 °C for 15 s and 72 °C for 30 s, 72 °C for 10min. PCR products of functional marker *WTK7-TM-FM* were mixed with 2 µl of loading buffer (98% formamide, 10mM EDTA, 0.25% bromophenol blue, and 0.25% xylene cyanol), separated on 1% agarose gels." We corrected it as suggested in Mat and Meth section lines 687-696.

Reviewer #2 (Remarks to the Author):

Li et al. reported the cloning of *Pm36*, a unique membrane associated tandem kinase (WTK7-TM), by combination of PacBio SMRT long-read genome sequencing and map-based cloning. The gene was validated by mutants and VIGS, and the proof is solid. *Pm36* could be only found in the wild emmer wheat populations and showed resistance to 104 *Bgt* isolate. *Pm36* has been introgressed into durum and common wheat background showing great gain in wheat resistance breeding.

Response: We would like to thank you for your careful reading, helpful comments, and constructive suggestions, which has significantly improved the presentation of our manuscript.

The authors did some mechanism research, but the story is not clear and insufficient. Some results are different with previous publication which should be carefully discussed or repeat to be suitable for publication. Please see 'Nirmala, J., Drader, T., Chen, X., Steffenson, B. and Kleinhofs, A., 2010. Stem rust spores elicit rapid RPG1 phosphorylation. Molecular plant-microbe interactions, 23(12), pp.1635-1642'. In this paper, RPG1 protein is phosphorylated within 5 min by exposure to spores from avirulent but not virulent races of stem rust. The pseudokinase pK1 domain is required for disease resistance but not phosphorylation.

Response: Thanks for your suggestions. In our study, we cloned the wheat *WTK7-TM* gene which encodes a tandem kinase protein including Kinase I (Kin I) and Kinase II (Kin II) domains, and a predicted single transmembrane domain in the C-terminus. We, therefore, explored *WTK7-TM* *in vitro* phosphorylation activity. The results showed that the full length *WTK7-TM*, Kin I or Kin II domain alone doesn't exhibit any auto-phosphorylation or catalytic activity, while the truncated version without the TM domain, MBP-*WTK-TM*^{Kin I-Kin II} does demonstrate such activities. (Figs. 5b, c). We guess the full length *WTK7-TM* phosphorylation is also required for recognition of *Blumeria graminis* f. sp. *tritici* (*Bgt*) *in vivo* which is similar with reported RPG1 phosphorylation depending on the recognition of *Puccinia graminis* f.sp. *tritici*. We will study the *in vivo* posttranslational modification of *WTK7-TM* in response to *Bgt* inoculation in future.

Questions/suggestions

1. Line 80: ref19: has been published on nature plants now, update the citation.

Response: Thanks. We corrected it as suggested in revised version.

2. What is the subcellular localization of *WTK-TM*, since it contains a transmembrane domain, did you have some proof as the TM domain located on the membrane as shown in Fig.1e?

Response: Thank you for your advices. To investigate the subcellular localization of *WTK7-TM*, we built recombinant plasmid *proUbi: WTK7-TM-GFP* to express *WTK7-TM* protein in wheat leaf mesophyll protoplasts. We found that *WTK7-TM* protein is localized in the nucleus, cytomembrane and cytoplasm in the wheat leaf mesophyll protoplasts (Supplementary Fig. 16). We have updated this information in the revised manuscript (Lines 266-268).

3. It is better to check ROS and cell death response of WTK7-TM. Is the intraROS play role for the WTK7-TM resistance? Follow this paper, <https://apsjournals.apsnet.org/doi/10.1094/PHYTO-07-22-0271-F1>

Response: Thank you for your advices. 3D232 showed highly resistance to *Bgt* E09 isolate accompanied by intracellular ROS production and cell death response, while Xueza0 exhibited a highly susceptible with producing fungal mycelium. The result was revised in the revision as suggested in Fig 1a.

4. Line 210 and 258: What is the unique resistance mechanism or different resistance mechanism for the WTK7-TM? If you do not have a clear proof, please tone down this part.

Response: Thanks for the comments and suggestions. We corrected the sentence as suggested in the revised manuscript (Line 265).

5. Have you checked the entire protein WTK7-TM has the autophosphorylation or not? I surmised the kinase domain alone may has autophosphorylation, the entire WTK7-TM may have kinase activity after reorganization with effector.

Response: Thanks for the comments and suggestions. We explored WTK7-TM *in vitro* phosphorylation activity. The results showed that the full length WTK7-TM, Kin I or Kin II domain alone doesn't exhibit any auto-phosphorylation or catalytic activity, while the truncated version without the TM domain, MBP-WTK-TM^{Kin I-Kin II} does demonstrate such activities (Fig. 5b, c and Supplementary Fig. 18). We agree with your speculation that the full length WTK7-TM phosphorylation is required for recognition of *Blumeria graminis* f. sp. *tritici* (*Bgt*) *in vivo* which is similar with reported RPG1 phosphorylation depending on the recognition of *Puccinia graminis* f.sp. *tritici*. We added the result in revised manuscript (Lines 379-386).

6. Have you check the different WTK7-TM alleles among the published wheat reference genomes? Are there some homologues, orthologs, paralogs? Could the insertion part sequence be original from other location of wheat chromosome, or original from other wheat relatives, or it is just totally a new sequence?

Response: Thanks for the suggestions. We performed a BLASTP analysis of the WTK7-TM protein in the published wheat and relative species reference genomes and did not find homologues, orthologs, paralogs of WTK7-TM protein. We think that this part of the insertion sequence may come from a closely related species, however, no evidence has been found.

7. Did you have some information (collect locations or wheat lines) on the 104 *Bgt* isolates, references or Table? Are they present as enough genetic diversity?

Response: Thank you for your advices. We added collect locations of 104 *Bgt* isolates and infection types of 3D232 in Supplementary Table 1. The 104 *Bgt* isolates were collected from 15 provinces in China, which represent enough genetic diversity.

8. All the figure legends of supplementary Figures should be improved. Such as, add

lactation information in Sup Fig.2. How did you do *Bgt* infection in field or just nature infection?

Response: Thank you for your advices. We revised the legends of supplementary Figures as you suggested. At the jointing stage, lines 3D232 and Xueza0 were inoculated with *Bgt* isolate E09 under field condition at the Gaoyi Experimental Station at Shijiazhuang in Hebei province, China. Powdery mildew reactions evaluated at the grain-filling stage were recorded as describe above. We have been added it in the revised manuscript (Lines 469-472 in Methods).

Add the gene information on supp Fig.3. and the main legend should be short no need to add the website address.

Response: Thanks for the suggestions. We have added the gene information and revised the sentence in new Supplementary Fig. 7.

The supp Fig.4 should be “Genes expression analysis in the MI3D232 genetic region.”

Response: Thank you for your advices. It has been corrected in new Supplementary Fig. 3.

Supp. Fig. 5 “transgenic complementation” may be used for overexpression functional genes in the recessive mutations.

Response: Thank you for your careful review. In order to validate the genes function of *CYB561-Domon* and *TM9SF4*, two overexpression constructs, *ProUbi:CYB561-Domon* and *ProUbi:TM9SF4*, driven by the maize (*Zea mays* L.) *Ubiquitin* promoter (Supplementary Fig. 6a, b), were separately delivered into the susceptible wheat cultivar Fielder by *Agrobacterium*-mediated transformation. Unfortunately, all plants positive for *ProUbi:CYB561-Domon* and *ProUbi:TM9SF4* constructs in the independent T₁ transgenic families were highly susceptible to the *Bgt* isolate E09 (Supplementary Fig. 6c, d), suggesting *CYB561-Domon* and *TM9SF4* are not the causal genes of *MI3D232*.

Supp Fig.6 - How do you know the original segment, duplication 1 and 2, and the MYA years of each duplication and each gene?

Response: Thank you for your questions. For the estimation of TEs' insertion time, first, we used Repeat Sequence Database (TREP) (<https://trep-db.uzh.ch/>) to identify known repetitive DNA. And then, the long terminal repeat sequences of LTR retrotransposable elements were delineated using Dotter analysis and manually checked. The evolutionary distance of the two LTR sequences was estimated by MEGA11 (<https://www.megasoftware.net/>) using a substitution rate of 1.3×10^{-8} mutations per site per year. For the duplication of the fragments, fragment 1 and fragment 2 are 92 kb and 69 kb, respectively. Except the large insertion/deletion (about 23 kb), fragment 1 and fragment 2 are only different in SNPs and small insertion/deletion. We aligned the sequences of fragment 1 and fragment 2 and then calculate their divergence time by MEGA11 as above mentioned. As for the original genomic region, there were only gene sequences (F-box and serpin) and a partial of repetitive sequences were conserved. We use these sequences to calculate their divergence time. However, It's hard to say which one is original and we only

can calculate their divergence time. The result was revised in the revision.

Supp. Fig.7. Which version of the reference are used? Add how do you do the analysis.

Response: Thank you for your questions. We use the reference genomes of the Chinese Spring (IWGSC RefSeq v1.0) and wild emmer genome assembly (Zavitan WEWSeq v.1.0). The genomic sequences were cut out using the flanking markers *WGGBM2* and *WGGBM7* on the *MI3D232/Pm36* locus based on Chinese Spring RefSeqv1.0 reference genome, Zavitan WEW_v1.0 reference genome and 5BIL-29 contig utg0040681. Sequence similarity comparison were performed using YASS genomic similarity search tool (<https://bioinfo.cristal.univ-lille.fr/yass/yass.php>). We have added the legend of Supplementary Fig. 8.

Supp Fig.9 add information what reference you used, the long-read assembly contigs?

Response: Thank you for your advices. We did the RNA-seq analysis using the 1.17 Mb physical interval of *Pm36* between the closest flanking markers *WGGBM2* and *WGGBM7* from long-read assembly contig ptg000422l.

Supp. Fig. 10. How do you calculate the different transcriptional isoforms?

Response: Thank you for your question. We choose one hundred single colonies in random and sequenced them using the Sanger sequencing. Then we can get the number of different transcriptional isoforms. We have revised the information in the Supplementary Fig. 10.

Supp. Fig 12 a, two should be Two.

Response: Thanks for your careful review. We have corrected it.

The title legend of Fig 13 is not good, and the protein structure is not clear, why you put the Rgp1 and Pm36 together?

Response: Thanks for the comments and suggestions. The legend of Supplementary Figure 13 as suggested has been improved in new Supplementary Figure 15. In this figure, we want to show the protein topology of the reported tandem kinase proteins, Yr15, Sr60, Pm24, WTK4, Sr62, Rpg1, and WTK7-TM, especially to illustrate the difference of transmembrane structure. We arranged the figures in the gene symbol order in barley Rpg1, wheat Yr15 (WTK1), Sr60 (WTK2), Pm24 (WTK3), WTK4, Sr62 (WTK5), Lr9 (WTK6-vWA) and Pm36 (WTK7-TM).

You need to add the PCR product size information in the Fig. 15, and you did not show how you develop the specific markers? What is the meaning of *Pm36* in the Fig. 15?

Response: Thanks for the suggestions. We have added the PCR product size information in the supplementary Fig. 19. Based on the genomic sequence of *WTK7-TM* gene, we developed a specific primer, *WTK7-TM-FM*, to identify the presence of the *WTK7-TM* functional allele among various *Triticeae* species. The dominant functional marker *WTK7-TM-FM* amplified a 1,198 bp PCR product if the plant carrying *WTK7-TM* gene, otherwise no amplification product. PCR products of functional marker *WTK7-TM-FM* were mixed

with 2 µl of loading buffer (98% formamide, 10mM EDTA, 0.25% bromophenol blue, and 0.25% xylene cyanol), separated on 1% agarose gels. We have changed the Pm36 to 5BIL-29 in the Supplementary Fig. 19.

You need to write where you get this map in Fig. 19? Google map or? Please consider to add the country name of Israel in the map since the most WEW is original from here.

Response: Thanks for the suggestions. We get this map from Prof. Lin Huang, Sichuan Agricultural University, China. Prof. Huang has used this map in the article entitled as “Distribution and Nucleotide Diversity of *Yr15* in Wild Emmer Populations and Chinese Wheat Germplasm”. We have added the country name of Israel in the map as you suggested.

Reference

He Y, Feng LH, Jiang Y, et al. Distribution and Nucleotide Diversity of *Yr15* in Wild Emmer Populations and Chinese Wheat Germplasm. *Pathogens* **9**, 212(2020).

9. Supp. Fig.11. I think it is better to add mock infection (non-infection) as control. WTK1 expression is reduced after *Pst* infection. While your result is different and need to be careful.

Response: Thanks for the suggestions. We have added mock infection as control in the experiment of *WTK7-TM* expression. We conducted the gene expression experiment again and found that *WTK7-TM* could be slightly affected after *Bgt* isolate E09 infection in lines 3D232 and 5BIL-29.

10. You need to give the Kinase I and II a family name as previous WTK genes, such as LRR-8B or WAK?

Response: Thank you for your advices. Based on Hidden Markov Model, the two *WTK7-TM* kinase domains belong to LRR_8B (cysteine-rich kinases) subfamily, like as Rpg1. We have added information in revised manuscript (Lines 272-273).

11. The supp. Table should be Excel format.

Response: Thanks for the suggestions. We have changed Supplementary Tables to Excel format.

Reviewer #3 (Remarks to the Author):

In their study the authors describe the identification and isolation of a novel powdery mildew resistance gene originating from wild emmer wheat. For that, they sequenced the carrier, a tetraploid WEW-durum introgression line. I will focus my review on the genomics work carried out in this study, as the main area of my expertise.

I have a few points that require clarification and/or some more work:

A.) The genome assembly statistics, contiguity in particular, look good but not great. I understand the genome assembly was left in scaffold state, which is reasonable for the main objectives of this study (identification of the resistance gene). Nevertheless, I see a missed opportunity here, too. Advancing the genome assembly to a pseudomolecule level would not only make this a much more valuable resource for downstream and follow-up analyses, but it would also set the genomic region harboring the novel resistance gene into a full genomic context. This could be of relevance for assessing gene regulation in a (epi-)genomic context, as well as chromosome organization. I would like to suggest the authors to consider doing Hi-C and bringing the assembly to a full pseudochromosome level, which could be performed with very reasonable efforts and costs.

Response: Thanks for the comments and suggestions. In order to advance the genome assembly of 5BIL, we generated an additional cell data of HiFi sequencing. Totally, PacBio CCS (HiFi) reads corresponding to approximately 19.43-fold coverage of tetraploid wheat genome were generated (Supplementary Table 3). As you suggested, we also performed Hi-C sequencing and generated about 117.31-fold Hi-C reads. After sequences assembly, 2,207 contigs were obtained, with a contig N50 of 20.8 Mb and N90 of 4.5 Mb, the longest contig length of 168.3 Mb, and the total length of the genome assembly of 10.5 Gb, which covered the whole genome of WEW Zavitan (~10.4 Gb) (Supplementary Table 3). The assembling values have a significant improvement from 15,347 to 2,207 contigs, and N50 from 3.2 to 20.8 Mb. However, we still haven't been able to assemble it at the full pseudochromosome level. Considering the current objective of identification of the resistance gene *Pm36* in this paper and limited funding of the project, we hope to apply extra fund to support to assemble in full pseudochromosome level in the future.

B.) While synteny/collinearity analyses have been performed with other genomes/species, I'm missing a proper comparison of the resistance carrying region with the homoelogenous region on the other WEW subgenome. Supplementary figure 3 needs a complete makeover in my view, I do not intuitively understand what is displayed here. It is also missing any explanatory figure legend.

Response: Thanks for the comments and suggestions. We provide a new Supplementary Figure 7 to replace the original Supplementary Figure 3. In this supplementary Figure, we displayed the highly conservation of the *M13D232/Pm36* genomic region on 5BL among multiple wheat reference genomes and the homoelogenous region of the WEW subgenome 5AL. No resistance genes were found in this region among currently reported wheat reference genomes except the tetraploid WEW-durum introgression line 5BIL-29 that carry

the *Pm36*. Compared with Zavitan, introgression line 5BIL-29 contains a 900 kb insertion harboring additional seven predicted genes with two segmental duplications in the *Pm36* physical region. The figure legend was also updated to explain the figure content. Please see the new Supplementary Fig. 7.

C.) Gene prediction and annotation: the methods detail that FgenesH and BlastP was used to identify genes in the respective genomic region. To understand what was done here, and to assess the prediction quality, this part needs more explanations and details. What database was blasted against? How were the final gene structures modeled? What was the impact of the transcriptome data on gene predictions? These are very critical points as many of the downstream analysis dealing with the novel resistance gene actually depend on this gene structure prediction.

Response: Thanks for your suggestions. For gene prediction and annotation, the 1.17 Mb genomic sequence, which can entirely cover the *MI3D232/Pm36* physical mapping interval from contig ptg000422I (7.1 Mb) was annotated using Softberry FGENESH and NCBI BLASTP

(https://blast.ncbi.nlm.nih.gov/Blast.cgi?PROGRAM=blastp&PAGE_TYPE=BlastSearch&LINK_LOC=blasthome) against nr Database to identify homologous proteins. Repetitive sequences were identified through BLASTN against the *Triticeae* repetitive elements (TREP) (<https://trep-db.uzh.ch/>). The long terminal repeat sequences of LTR retrotransposable elements were delineated using Dotter analysis and manually checked. RNA-Seq data was also used to predict the gene structure. Among the eleven annotated genes in the 1.17 Mb genomic interval of the *MI3D232/Pm36* locus, only *CYB561-Domon*, *TM9SF4*, and *WTK7-TM* were expressed, while the three *F-box* and five *serpin* genes were not expressed in the leaves of 5BIL-29 after inoculation with *Bgt* isolate E09 (Supplementary Table 5).

D.) The authors describe transcriptional variants of the resistance gene (Supp Figure 10), but I'm not clear where these actually come from and what the supporting evidence for that distribution is...did the authors also utilize long-read expression data (Iso-Seq) to validate those structures?

Response: Thank you for your advices. The full-length cDNAs of *WTK7-TM* were also obtained from lines 3D232 by the 5'- and 3'-rapid amplification of complementary DNA (cDNA) ends (RACE). Nine transcription isoforms of *WTK7-TM*, named as *TV1-TV9*, were identified, with isoform *TV1* being the most common among them (Supplementary Fig. 10). We have also added the proportion of each transcript in Supplementary Fig.10. One hundred colonies were sequenced using the Sanger sequencing. The numbers and percentages of variants are shown on the right, e.g., 59/100 (59%). The *WTK7-TM* gene has 12 exons, which encodes a 667-residue tandem kinase protein including Kinase I (Kin I) and Kinase II (Kin II) domains, and a single transmembrane domain in the C-terminus (Fig. 1e and Supplementary Fig. 11). We have added information in revised manuscript (Lines 199-206).

E.) Supp Figure 9: it is hard to see differences in this screenshot, and a figure legend is

missing as well. I suggest to re-do this figure.

Response: Thanks for the comments and suggestions. To avoid confusion, we deleted the Supplementary Fig.9 and added the Supplementary Table 5 to display RNA-seq analysis results. Among the eleven annotated genes in the 1.17 Mb genomic interval of the *MI3D232/Pm36* locus, only *CYB561-Domon*, *TM9SF4*, and *WTK7-TM* were expressed, while the three *F-box* and five *serpin* genes were not expressed in the leaves of 5BIL-29 after inoculation with *Bgt* isolate E09 (Supplementary Table 5).

Reviewers' Comments:

Reviewer #1:

Remarks to the Author:

You responded to most of the comments but I still have few concerns.

The discussion about using resistant genes to prevent resistance development has not been updated.

In your response to the reviewers, you mentioned that Pm21 has been extensively utilized in China.

However, do you have any information on Pm36? Providing information about Pm36 would enable the reader to formulate hypotheses about broad-spectrum resistance. I was unable to locate the reference 'Chen et al. 2013.

The origin of the different wheat accessions are provided in the PDF file in response to reviewers' comments but it has not been included in the manuscript.

L134 "around" is not necessary.

L223 I would say necessary instead of "responsible".

L258 this is figure 3 and not 2.

Figure 3 please provide the number of leaves used per biological replicate and the method used to calculate the relative expression. How many T1 plants were evaluated per family? It would have been great to see the response of a family that followed the transformation process but without the transgene.

L267 According to Supplementary Figure 16, it is quite challenging to determine whether the WTK7 protein is localized in the cytoplasm, the plasma membrane, or both. To discern its exact subcellular localization, it may be necessary to induce plasmolysis in the cells or employ labeled proteins with known subcellular localizations. Furthermore, it is important to note that there might be some free GFP expressed in the protoplast with WTK7, which could potentially account for the observed cytoplasmic localization, if indeed present. Notably, the signal in WTK7 protoplasts appears distinct from that in the GFP protoplasts.

L286, please provide in the text what is truncated and in the corresponding figure, I guess it would be great to include Phosphorylated myelin basic protein instead of Myelin basic protein.

L297 this is mentioned that the 38 WEW shared an identical genomic sequence and CDS. This means that you have sequenced this gene from the 38 WEW. Is it right? as it is not provided in the material and method section.

L699 This is ST2 and not ST1.

In the material and method section, I'm not sure to get it right one biological sample = one leave?

And you've done three independent experiments (this is what you called biological replicates?) because biological replicates is used for different meaning here.

Supplementary Figure 12. What are the size of the bands of the DNA ladder? I do not see any detail in the material and method section regarding this experiment, the number of roots, replicates, how the RT was done, the method for the relative expression...

SM19 What are the size of the bands of the ladder?

Reviewer #2:

Remarks to the Author:

The authors made a lot of efforts to improve the manuscript, it now looks almost ready for publication.

I only have one question/concern; to prove the localization, the authors should do a western blot.

It is possible that the WTK7-TM-GFP can get cleavage and just GFP will go to the nucleus and cytoplasm.

This happens a lot when they are overexpressed.

The authors should show no cleavage of the protein to say anything. It could be a mix (full length at the membrane, cleaved versions elsewhere).

Other than this, I am satisfied from this MS.

Reviewer #3:

Remarks to the Author:

I appreciate the modifications implemented by the authors in this round of revision. Supp Figure 7 now is much more intuitive. Related to this analysis, did the authors compare the marker-delineated regions based on the annotated genes only, or also on a genome sequence level? This question refers to the differences in gene annotation on independent assemblies, as orthologous genes might just not have been annotated or classified as Low-Confidence genes or TEs.

I understand that funding and manpower is limited for generating a reference-quality genome assembly, but it would massively increase its value for downstream analyses. I do not think scaffold level assemblies would be overly useful nowadays, other than for the intended use in this study.

Point-by-point response to Reviewers

REVIEWER COMMENTS

Reviewer #1 (Remarks to the Author):

You responded to most of the comments but I still have few concerns.

Response: We sincerely thank the reviewer for the positive evaluation and cogent comments to help us improve our manuscript.

The discussion about using resistant genes to prevent resistance development has not been updated. In your response to the reviewers, you mentioned that Pm21 has been extensively utilized in China. However, do you have any information on Pm36? Providing information about Pm36 would enable the reader to formulate hypotheses about broad-spectrum resistance. I was unable to locate the reference 'Chen et al. 2013.

Response: We thank the reviewer for this suggestion. We revised the discussion section about using resistant genes to prevent resistance development (lines 428-443 in Discussion). “The powdery mildew resistance gene *Pm36* was transferred to common wheat line 3D232 from wild emmer accession I222 in 2003, and it is still one of the most effective resistance genes in China. Wheat line 3D232 showed broad-spectrum resistance against genetically diverse *Bgt* isolates (Supplementary Fig. 1 and Supplementary Table 1) without a growth penalty. Yet, *Bgt*-resistant wheat cultivars carrying single race specific *Pm* gene are prone to lose their resistance due to strong pathogen selective pressure. Therefore, combining multiple and diverse resistance genes is a sustainable approach to disease management in wheat breeding.” We added the discussion about *Pm36* gene. Please find the reference “Chen et al. 2013” below.

Reference

Chen, P. *et al.* Radiation-induced translocations with reduced *Haynaldia villosa* chromatin at the *Pm21* locus for powdery mildew resistance in wheat. *Mol. Breeding* 31, 477–484 (2013).

The origins of the different wheat accessions are provided in the PDF file in response to reviewers' comments but it has not been included in the manuscript.

Response: We thank the reviewer for pointing out the issue. We added the origin of the different wheat accessions as suggested in Mat and Meth section.

Most of these accessions were obtained from the Wheat Genetic and Genomic Resources Center, Kansas State University, USA, The USDA-ARS National Small Grains Collection, USA, and University of Haifa, Israel. The 262 entries of Chinese wheat mini-core collection (MCC) were available in National Key Facility for Crop Gene Resources and Genetic Improvement, Chinese Academy of Agriculture Sciences. The 125 cultivated emmer wheat (*T. turgidum* subsp. *dicoccum*), 230 durum (*T. turgidum* subsp. *durum*) has the official Plant ID, such as PI ×××, CItr ××× as shown in Supplementary Table 7.

Please refer to lines 470-477 for details.

L134 “around” is not necessary.

Response: We deleted the “around” word as suggested.

L223 I would say necessary instead of “responsible”.

Response: We thank the reviewer for the suggestion. Revised accordingly.

L258 this is figure 3 and not 2.

Response: Thanks for picking this up this error, it has been revised accordingly.

Figure 3 please provide the number of leaves used per biological replicate and the method used to calculate the relative expression. How many T₁ plants were evaluated per family? It would have been great to see the response of a family that followed the transformation process but without the transgene.

Response: Thank you so much for the careful review. In Figure 3c, *WTK7-TM* of expression values are means ± SD from three biological replicates (three leaves used per biological replicate) and three technical replicates. In our study, 15 individuals of T₁ transgenic families were inoculated with *Bgt* isolate E09. We have added the information in Figure 3 and Mat and Meth section (lines 663-669). We have added infection types of two negative transgenic lines (without the *WTK7-TM* transgene) *WTK7-TM-COM6* and *WTK7-TM-COM7* in response to *Bgt* isolate E09 as suggested in Figure 3.

L267 According to Supplementary Figure 16, it is quite challenging to determine whether the WTK7 protein is localized in the cytoplasm, the plasma membrane, or both. To discern its exact subcellular localization, it may be necessary to induce plasmolysis in the cells or employ labeled proteins with known subcellular localizations. Furthermore, it is important to note that there might be some free GFP expressed in the protoplast with WTK7, which could potentially account for the observed cytoplasmic localization, if indeed present. Notably, the signal in WTK7 protoplasts appears distinct from that in the GFP protoplasts.

Response: Thank you for your insightful comment. We performed additional experiment to confirm the result. Wheat protoplasts were transfected with the ProUbi::WTK7-TM-GFP plasmid. Then total protein extracted from wheat protoplasts were separated by SDS-PAGE for Western blot analysis. The result showed that GFP band and the WTK7-TM-GFP band were detected in the samples (ProUbi::WTK7-TM-GFP) comparing only GFP band in the control (ProUbi::GFP). Therefore, we hypothesized that whether WTK7-TM+GFP fusion proteins undergo breakage during the sample preparation process. Consequently, we gently lysed the protein with IP buffer and denatured at 42°C for 2 hours. However, we also observed the GFP band and WTK7-TM-GFP band in the samples (ProUbi::WTK7-TM-GFP), which is similar to those denatured under the 100°C 10 min. These results suggested that some free GFP generated from WTK7-TM-GFP protein cleavage might expressed in the protoplast with WTK7-TM. Then we investigated the subcellular localization pattern of WTK7-TM in *N. benthamiana* leaves. 35S::WTK7-TM-GFP+35S::NAA60-mCherry or 35S::GFP+35S::NAA60-mCherry fusion proteins were expressed in *N. Benthamian* leaves through agroinfiltration. NAA60 was used as plasma membrane marker protein (Linster et al., 2020). WTK7-TM was mainly localized in plasma membrane as demonstrated by co-localization with NAA60 as well as western blot assays. Although WTK7-TM has a low expression, we only observed WTK7-TM-GFP band in 35S::WTK7-TM-GFP+35S::NAA60-mCherry samples and GFP band in 35S::GFP+35S::NAA60-mCherry samples.

Supplementary Fig Subcellular localization of WTK7-TM in wheat protoplasts.

Reference

Linster, E. et al. The Arabidopsis N^α-acetyltransferase NAA60 locates to the plasma membrane and is vital for the high salt stress response. *New Phytol.* 228, 554-569 (2020).

L286, please provide in the text what is truncated and in the corresponding figure, I guess it would be great to include Phosphorylated myelin basic protein instead of Myelin basic protein.

Response: We thank the reviewer for pointing out these issues. We revised the sentence to “The results showed that the full length WTK7-TM (1-667 aa), Kin I (1-320 aa) or Kin II (321-646 aa) domain alone doesn’t exhibit any auto-phosphorylation or catalytic activity, while the truncated version WTK-TM^{Kin I-Kin II} (1-646 aa) composing of Kin I+ Kin II without the transmembrane domain does demonstrate such activities (Fig. 5b, c and Supplementary Fig. 18)” in the lines 287-290. We also added detailed descriptions of different truncated versions in the legend of Fig 5 and Supplementary Fig. 18. We replaced “myelin basic protein” with “Phosphorylated myelin basic protein” in Fig 5c and Supplementary Fig. 18 according to the suggestion.

L297 this is mentioned that the 38 WEW shared an identical genomic sequence and CDS. This means that you have sequenced this gene from the 38 WEW. Is it right? as it is not provided in the material and method section.

Response: Thanks for picking this up. We added description “The coding sequence (CDS) of the WTK7-TM gene were amplified from 38 WEW accessions, which were tested carrying WTK7-TM gene by functional marker WTK7-TM-FM, using the primers listed in Supplementary Table 2. ClustalW was used for multiple sequence alignment.” in material and method section according to the suggestions. Please refer to lines 735-739 for details.

L699 This is ST2 and not ST1.

Response: We revised “Supplementary Table 1” to “Supplementary Table 2”, please refer to line 742.

In the material and method section, I’m not sure to get it right one biological sample = one leave? And you’ve done three independent experiments (this is what you called biological replicates?) because biological replicates is used for different meaning here.

Response: Thanks for the comments. We apologize for the unclear description. In this study, three biological replicates were performed for quantitative real-time PCR (qRT-PCR). Three leaves were

collected and analyzed per biological replicate. Three technical replicates for each leaf were also conducted. We have revised in Mat and Meth section.

Supplementary Figure 12. What are the size of the bands of the DNA ladder? I do not see any detail in the material and method section regarding this experiment, the number of roots, replicates, how the RT was done, the method for the relative expression...

Response: We thank the reviewer for pointing out these issues. We added the size of the bands of the DNA ladder in Supplementary Figure 12a. To investigate whether *WTK7-TM* is expressed in different wheat tissues, we conducted RT-PCR analysis in leaf, stem, and root of 3D232 at the two-leaf stage. *TaActin* was used as the endogenous control. Three biological replicates (three leaves or root or stem used per biological replicate) and three technical replicates for each sample were performed. Different replicates have similar results. we added a description of RT-PCR analysis in in Mat and Meth section (Lines 587-592).

SM19 What are the size of the bands of the ladder?

Response: We thank the reviewer for pointing out the issue. In order not to cause ambiguity, we revised Supplementary Figure 19 and added the size of the bands of the ladder.

Reviewer #2 (Remarks to the Author):

The authors made a lot of efforts to improve the manuscript, it now looks almost ready for publication.

Response: We thank the reviewer for the positive comments on our work.

I only have one question/concern; to prove the localization, the authors should do a western blot. It is possible that the WTK7-TM-GFP can get cleavage and just GFP will go to the nucleus and cytoplasm. This happens a lot when they are overexpressed. The authors should show no cleavage of the protein to say anything. It could be a mix (full length at the membrane, cleaved versions elsewhere). Other than this, I am satisfied from this MS.

Response: Thank you for your valuable comments and good suggestions. Wheat protoplasts were transfected with the ProUbi::WTK7-TM-GFP plasmid. Then total protein extracted from wheat protoplasts were separated by SDS-PAGE for Western blot analysis. The result showed that GFP band and the WTK7-TM-GFP band were detected in the samples (ProUbi::WTK7-TM-GFP) comparing only GFP band in the control (ProUbi::GFP). Therefore, we hypothesized that whether WTK7-TM+GFP fusion proteins undergo breakage during the sample preparation process. Consequently, we gently lysed the protein with IP buffer and denatured at 42°C for 2 hours. However, we also observed the GFP band and WTK7-TM-GFP band in the samples (ProUbi::WTK7-TM-GFP), which is similar to those denatured under the 100°C 10 min. These results suggested that some free GFP generated from WTK7-TM-GFP protein cleavage might expressed in the protoplast with WTK7-TM. Then we investigated the subcellular localization pattern of WTK7-TM in *N. benthamiana* leaves. 35S::WTK7-TM-GFP+35S::NAA60-mCherry or 35S::GFP+35S::NAA60-mCherry fusion proteins were expressed in *N. Benthamian* leaves through agroinfiltration. NAA60 was used as plasma membrane marker protein (Linster et al., 2020). WTK7-TM was mainly localized in plasma membrane as demonstrated by co-localization with NAA60 as well as western

blot assays. Although WTK7-TM has a low expression, we only observed WTK7-TM-GFP band in 35S::WTK7-TM-GFP+35S::NAA60-mCherry samples and GFP band in 35S::GFP+35S::NAA60-mCherry samples.

Supplementary Fig Subcellular localization of WTK7-TM in wheat protoplasts.

Reference

Linster, E. et al. The Arabidopsis N^α-acetyltransferase NAA60 locates to the plasma membrane and is vital for the high salt stress response. *New Phytol.* 228, 554-569 (2020).

Reviewer #3 (Remarks to the Author):

I appreciate the modifications implemented by the authors in this round of revision.

Response: Thanks for the encouraging comments on our work.

Supp Figure 7 now is much more intuitive. Related to this analysis, did the authors compare the marker-delineated regions based on the annotated genes only, or also on a genome sequence level? This question refers to the differences in gene annotation on independent assemblies, as orthologous genes might just not have been annotated or classified as Low-Confidence genes or TEs.

Response: Thank you for your detailed comments and questions. In this study, micro-collinearity analysis on *MI3D232/Pm36* locus among *Triticeae* reference genome was performed at the annotated gene level. To avoid missing low-confidence genes, we masked low complexity genomic sequence using exemplar transposon sequences using RepeatMasker (<https://www.repeatmasker.org/>) and conducted a gene annotation analysis using Softberry FGENESH and NCBI BLASTP (https://blast.ncbi.nlm.nih.gov/Blast.cgi?PROGRAM=blastp&PAGE_TYPE=BlastSearch&LINK_LOC=blasthome) against nr Database to identify homologous proteins. Gene annotation results demonstrated that *serpin* genes showed variation in copy number and *peroxidase* genes, likely low-confidence genes, were annotated. While *F-box* gene annotated on *MI3D232/Pm36* locus in introgression line 5BIL-29 was absent in the its corresponding regions among remaining *Triticeae* genome. We revised Supp Figure 7 as suggested.

I understand that funding and manpower is limited for generating a reference-quality genome assembly, but it would massively increase its value for downstream analyses. I do not think scaffold level assemblies would be overly useful nowadays, other than for the intended use in this study.

Response: Thank you for your insightful suggestions. We agree with the reviewer's comments. A pseudomolecule-level genome assembly for tetraploid WEW-durum introgression line 5BIL-29 will

contribute to understanding the evolution of its genes and downstream and follow-up analyses. We would like to assemble a chromosome-level genome of 5BIL-29 in future initiatives.

Reviewers' Comments:

Reviewer #1:

Remarks to the Author:

Authors have addressed most of the comments.

I'm still not able to see reference Chen 2013 in the references section.

The supplementary table 7 is empty?

L267 there is a typo *N. benthamiana* (please edit also in the legend in of Supplementary Figure 16)

The subcellular localization has been conducted under improved conditions. However, it appears that GFP alone exhibits better colocalization with your plasma membrane control compared to the WTK7-TM-GFP protein. Do you have images where the plasma membrane is slightly detached from the cell wall to observe how the signals of the different proteins behave?

I still have questions about the replicates. I understand that one biological replicate = 3 leaves and three biological replicates were used to perform qRT-PCR. Could you please specify whether these 3 biological replicates were collected from the same experiment or during independent experiments.

Reviewer #2:

Remarks to the Author:

As far as my concern, the paper is now ready for publication, good work!

Reviewer #3:

Remarks to the Author:

The authors addressed most of my pending concerns.

My main point with Supp Figure 7 was that it should be checked and confirmed whether the F-box genes and WTK7 (identified only in 5BIL-29) are indeed absent in all other assemblies, or if they were missed or classified as low-confidence by a differing annotation approach. A simple BLAST search for these genes in the respective genome sequences would do this job.

Point-by-point response to Reviewers

REVIEWER COMMENTS

Reviewer #1 (Remarks to the Author):

Authors have addressed most of the comments.

Response: We appreciate your constructive comments and suggestions for improving the manuscript.

I'm still not able to see reference Chen 2013 in the references section.

Response: Revised accordingly. We added the reference Chen 2013 in the references section (Lines 933-935).

Reference

52. Chen, P. et al. Radiation-induced translocations with reduced *Haynaldia villosa* chromatin at the *Pm21* locus for powdery mildew resistance in wheat. *Mol. Breeding* **31**, 477–484 (2013).

The supplementary table 7 is empty?

Response: Supplementary table 7 described the distribution of WTK7-TM in tetraploid and hexaploid wheat natural populations. Since Supplementary Table 7 contains too much data, we provide a separate file in addition to the other Supplementary Tables.

L267 there is a typo *N. benthamiana* (please edit also in the legend in of Supplementary Figure 16).

Response: Sorry for this error. Revised accordingly in both the text (Line 267) and the legend of Supplementary Figure 16.

The subcellular localization has been conducted under improved conditions. However, it appears that GFP alone exhibits better colocalization with your plasma membrane control compared to the WTK7-TM-GFP protein. Do you have images where the plasma membrane is slightly detached from the cell wall to observe how the signals of the different proteins behave?

Response: Thank you for raising this point. For subcellular localization, fusion proteins 35S::WTK7-TM-GFP+35S::NAA60-mCherry or 35S::GFP+35S::NAA60-mCherry were expressed in *Nicotiana Benthamiana* leaves through agroinfiltration. For plasmolysis experiments, *N. Benthamiana* leaf samples were incubated 10-15 min in 4% NaCl before imaging as described Kolodziej et al (2021). The green fluorescence of WTK7-TM-GFP overlapped with the red fluorescence of NAA60-mCherry when the cytoplasm detached from the cell wall, indicating that WTK7-TM predominantly localized to the plasma membrane (Supplementary Fig. 16). We added a description of plasmolysis experiments in Mat and Meth section (Lines 681-682) and in the legend in of Supplementary Figure 16.

Reference

45. Kolodziej, M. C. et al. A membrane-bound ankyrin repeat protein confers race-specific leaf rust disease resistance in wheat. *Nat. Commun.* **12**, 956 (2021).

I still have questions about the replicates. I understand that one biological replicate = 3 leaves and three biological replicates were used to perform qRT-PCR. Could you please specify whether these 3 biological replicates were collected from the same experiment or during independent experiments.

Response: We apologize for any confusion regarding the replicates. In our study, three biological replicates were performed from three independent experiments. We have made revisions in Mat and Meth (Line 587).

Reviewer #2 (Remarks to the Author):

As far as my concern, the paper is now ready for publication, good work!

Response: Thanks for the encouraging comments on our work.

Reviewer #3 (Remarks to the Author):

The authors addressed most of my pending concerns.

Response: We thank the reviewer for the positive comments on our work.

My main point with Supp Figure 7 was that it should be checked and confirmed whether the F-box genes and WTK7 (identified only in 5BIL-29) are indeed absent in all other assemblies, or if they were missed or classified as low-confidence by a differing annotation approach. A simple BLAST search for these genes in the respective genome sequences would do this job.

Response: Thank you for your suggestion. We use the same pipeline to annotation the genes from different genome regions. In addition, we do the BLAST search with the parameter (1e-10) by using three *F-box* genes and also *WTK7-TM* gene against the available genome assembly, especially the corresponding regions of different genomes. We didn't find the corresponding three F-box genes and *WTK7-TM* gene in all other genome assemblies.

The detailed information of the corresponding genomes physical intervals are as following:

chr5B_ae.speltoides.AEG-9674-1:483410000-483630000
chr5B_ae.speltoides.TS01:412420000-412580000
chr5B_Wild_emmer:547540000-547810000
chr5B_Durum_Wheat:538440000-538720000
chr5B_Triticum_spelta_PI190962:541470000-541740000
chr5B_Renan:536980000-537160000
chr5B_Kariega:562040000-562350000
chr5B_Norin61:544900000-545170000
chr5B_Mace:528270000-528500000
chr5B_LongReach_Lancer:539760000-540030000
chr5B_CDC_Stanley:546230000-546550000
chr7B_ArinaLrFor:809300000-809560000
chr5B_Jagger:533560000-533840000
chr5B_CDC_Landmark:542090000-542360000
chr5B_Julius:555910000-556180000
chr5B_Fielder:555910000-556180000
chr5B_Chinese_Spring2.1:544460000-544730000.

Sequences can be download from <http://202.194.139.32/getfasta/index.html>

Reviewers' Comments:

Reviewer #1:

Remarks to the Author:

Thank you for these revisions. The subcellular localization looks much better now. I just have two minor comments. The new table S7 is named Table S6 and in *N. benthamiana*, the b should be in lowercase.

Reviewer #3:

Remarks to the Author:

The authors sufficiently addressed all my points raised.

Point-by-point response to Reviewers

REVIEWER COMMENTS

Reviewer #1 (Remarks to the Author):

Thank you for these revisions. The subcellular localization looks much better now.

Response: Thank you for the constructive suggestions and professional comments.

I just have two minor comments. The new table S7 is named Table S6 and in *N. benthamiana*, the b should be in lowercase.

Response: We appreciate your detailed comment. We have revised “Table S6” as “Supplementary Table 7”. We have revised “*N. Benthamiana*” to “*N. benthamiana*” in both the text (Lines 267 and 667) and the legend of Supplementary Figure 16.

Reviewer #3 (Remarks to the Author):

The authors sufficiently addressed all my points raised.

Response: Thank you very much for your valuable comments and constructive suggestions.